# Synthesis and Pro-Apoptotic Effects of Nitrovinylanthracenes and Related Compounds in Chronic Lymphocytic Leukaemia (CLL) and Burkitt’s Lymphoma (BL)

**DOI:** 10.3390/molecules28248095

**Published:** 2023-12-14

**Authors:** Andrew J. Byrne, Sandra A. Bright, James. P. McKeown, Adam Bergin, Brendan Twamley, Anthony M. McElligott, Sara Noorani, Shubhangi Kandwal, Darren Fayne, Niamh M. O’Boyle, D. Clive Williams, Mary J. Meegan

**Affiliations:** 1School of Pharmacy and Pharmaceutical Sciences, Trinity Biomedical Sciences Institute, Trinity College Dublin, 152-160 Pearse St, Dublin 2, D02 R590 Dublin, Irelandmckeowjp@tcd.ie (J.P.M.); mmeegan@tcd.ie (M.J.M.); 2School of Biochemistry and Immunology, Trinity Biomedical Sciences Institute, Trinity College Dublin, 152-160 Pearse St, Dublin 2, D02 R590 Dublin, Irelandkandwals@tcd.ie (S.K.); fayned@tcd.ie (D.F.); clive.williams@tcd.ie (D.C.W.); 3School of Chemistry, Trinity Biomedical Sciences Institute, Trinity College Dublin, 152-160 Pearse St, Dublin 2, D02 R590 Dublin, Ireland; twamleyb@tcd.ie; 4Discipline of Haematology, School of Medicine, Trinity Translational Medicine Institute, St. James’s Hospital and Trinity College, Dublin 8, D08 W9RT Dublin, Ireland; tony.mcelligott@tcd.ie

**Keywords:** nitrostyrene, anthracene, chronic lymphocytic leukaemia (CLL), Burkitt’s lymphoma (BL), antiproliferative

## Abstract

Chronic lymphocytic leukaemia (CLL) is a malignancy of the immune B lymphocyte cells and is the most common leukaemia diagnosed in developed countries. In this paper, we report the synthesis and antiproliferative effects of a series of (*E*)-9-(2-nitrovinyl)anthracenes and related nitrostyrene compounds in CLL cell lines and also in Burkitt’s lymphoma (BL) cell lines, a rare form of non-Hodgkin’s immune B-cell lymphoma. The nitrostyrene scaffold was identified as a lead structure for the development of effective compounds targeting BL and CLL. The series of structurally diverse nitrostyrenes was synthesised via Henry–Knoevenagel condensation reactions. Single-crystal X-ray analysis confirmed the structure of (*E*)-9-chloro-10-(2-nitrobut-1-en-1-yl)anthracene (**19f**) and the related 4-(anthracen-9-yl)-1*H*-1,2,3-triazole (**30a**). The (*E*)-9-(2-nitrovinyl)anthracenes **19a, 19g** and **19i–19m** were found to elicit potent antiproliferative effects in both BL cell lines EBV^−^MUTU-1 (chemosensitive) and EBV^+^ DG-75 (chemoresistant) with >90% inhibition at 10 μM. Selected (*E*)-9-(2-nitrovinyl)anthracenes demonstrated potent antiproliferative activity in CLL cell lines, with IC_50_ values of 0.17 μM (HG-3) and 1.3 μM (PGA-1) for compound **19g**. The pro-apoptotic effects of the most potent compounds **19a**, **19g**, **19i**, **19l** and **19m** were demonstrated in both CLL cell lines HG-3 and PGA-1. The (*E*)-nitrostyrene and (*E*)-9-(2-nitrovinyl)anthracene series of compounds offer potential for further development as novel chemotherapeutics for CLL.

## 1. Introduction

Chronic lymphocytic leukaemia (CLL) is a B-cell malignancy and is the most common form of leukaemia of the adult population in developed countries, accounting for 25% of all cases of leukaemia and 1.3% of all cancers [1]. CLL is largely a disease of the elderly [2,3], with greater than 70% of patients aged over 65 years at time of clinical diagnosis. However, the disease is now increasingly common in younger patients [4]. Ireland (along with Australia, Italy and the US) shows the highest incidence rates for CLL worldwide [5] of approximately 4.5 per 100,000 in males and 2 per 100,000 in females [6,7,8]. CLL is a clinically heterogeneous lymphoproliferative disorder characterised by the clonal expansion of CD5^+^ mature B-lymphocytes, usually involving the bone marrow, spleen, lymph nodes and peripheral blood [9,10,11]. The disease is clinically classified according to the mutational status of the immunoglobulin heavy chain gene (IGVH); CLL patients with mutated IGVH (M-IGVH) usually have an indolent form of the disease, while unmutated IGVH (UM-IGVH) is associated with a more aggressive disease course [12]. The confirmation of a clonal population of B cells greater than 5000/mL of blood is diagnostic in most cases of CLL together with the expression of surface CD5 and CD23. Genomic and molecular markers are useful in assessing CLL prognosis; TP53 mutations, UM-IGVH, del(17p) and del(11q), together with complex karyotype, are associated with a poor prognosis. A favorable CLL prognosis is associated with del(13q), while normal karyotype and trisomy 12 are regarded as intermediate prognostic factors for CLL. A number of emerging prognostic markers for CLL are now identified, including mutations in Notch receptor 1 (NOTCH1), splicing factor 3B subunit 1 (SF3B1), baculoviral IAP repeat-containing 3 (BIRC3) and ATM serine/threonine kinase (ATM) [13,14,15].

Chemotherapeutic drugs used for CLL include the cytostatic nucleoside prodrug fludarabine phosphate **1** (Figure 1) and the alkylating agents cyclophosphamide, chlorambucil and bendamustine [16]; however, a number of very effective oral targeted therapies (such as ibrutinib **2**, idelalisib **3** and venetoclax **4**) are now available. Ibrutinib **2**, a Bruton’s tyrosine kinase (BTK) inhibitor approved for CLL [17], forms a covalent bond with cysteine residue Cys481 at the ATP binding site of BTK. Idelalisib **3** is a phosphoinositide 3-kinase δ (PI3Kδ) inhibitor, which inhibits B-cell receptor signalling and is approved for the treatment of relapsed CLL [18] and mantle cell lymphoma [19]. Venetoclax **4** is a highly potent, orally bioavailable selective inhibitor of the anti-apoptotic B-cell lymphoma-2 protein (Bcl-2) [20]. Many non-covalent BTK inhibitors have been reported to be effective in CLL and multiple other B-cell malignancies, e.g., GDC-0853 [21] and pirtobrutinib **5** [22]. The second-generation BTK inhibitor acalabrutinib **6** forms a covalent bond with the key cysteine residue (Cys481) of BTK, resulting in inhibition of BTK. [23] Zanubrutinib **7**, a next-generation BTK inhibitor, was approved by the FDA in 2023 for treatment of patients with CLL or small lymphocytic lymphoma (SLL) [24]. Immunotherapies such as the anti-CD20 monoclonal antibodies rituximab, obinutuzumab, ofatumumab and the anti-CD52 antibody alemtuzumab have proven successful in treating CLL [16,25,26]. Richter’s transformation (RT) of CLL to an aggressive B-cell lymphoma is a complication of CLL; however, the introduction of the PD-1-blocking antibodies pembrolizumab and nivolumab [27] show selective efficacy in CLL patients with RT [28].

Despite the recent advancements in the targeted clinical treatment of CLL, there still remains an urgent requirement for the discovery and development of novel therapeutic agents to combat acquired disease resistance as opposed to maintenance alone [29]. Examples of preclinical development of CLL-directed small-molecule therapies include the novel tubulin-targeting agent pyrrolo-1,5-benzoxazepine-15 (PBOX-15), which induces apoptosis in poor prognostic subgroups of CLL patients [30], while 25-hydroxyvitamin d-24-hydroxylase (CYP24A1) inhibitors have demonstrated efficacy in primary chronic CLL cells [31].

Burkitt’s lymphoma (BL) is a rare, aggressive non-Hodgkin’s lymphoma (NHL) that affects B-lymphocytes. The endemic form has a high incidence in equatorial Africa [32], whereas the sporadic form is identified in 1–2% of adult lymphomas globally and up to 40% of pediatric lymphomas in US and western Europe [33]. Treatments for BL include a combination of rituximab with chemotherapeutics such as vincristine, methotrexate, doxorubicin and cyclophosphamide [34,35]. With increased reports of immunodeficient HIV-linked BL [36] and the association of endemic BL with Epstein–Barr virus (EBV) [37], selective and potent treatments for BL are required as prognosis for relapsed BL is extremely poor [38].

Previous studies by our research group have shown that nitrostyrenes [39] and related (*E*)-9-(2-nitrovinyl)-9,10-dihydro-9,10-ethanoanthracene compounds such as **8** (designed from the tetracyclic scaffold structure of the antidepressant maprotiline **9**) [40] have potent antiproliferative and pro-apoptotic effects in BL cell lines MUTU-I (chemosensitive) and DG-75 (chemoresistant), as shown in Figure 2 [41]. These compounds were initially investigated in relation to the role of serotonin transporter (SERT) in B-cell malignancies including BL and demonstrated the anti-proliferative and pro-apoptotic effects of nitrostyrene amphetamine-related compounds [42,43,44].

The nitrostyrene-containing compounds such as **10a–c** reduced cell viability effectively in both BL and CLL cell lines and were superior to the clinical drugs fludarabine phosphate and taxol [39,45]. The IC_50_ values in the CLL cell lines were in the low-micromolar range (2–5 μM) irrespective of IGVH mutational status (I83 and PGA-1: mutated-IGVH; HG-3 and CII: unmutated-IGVH). This result compared favourably with IC_50_ values of 20–50 μM obtained for fludarabine phosphate (a current CLL frontline treatment) in these cell lines. We also identified the related (*E*)-9-(2-nitrovinyl)anthracene **19a** as a potent antiproliferative agent in the BL cell lines MUTU-I (IC_50_ = 3 µM) and DG-75 (IC_50_ = 1.5 µM), which induced apoptosis in both BL cell lines [41,42]. The (*E*)-(2-nitrovinyl)benzene pharmacophore was thus identified as a scaffold that has demonstrated relevant anticancer activity [45]. The biological macromolecular target(s) of these compounds, driving the antiproliferative response, is as yet unknown, so we focussed our work on phenotypic cellular responses. We wished to investigate the preclinical potential of a panel of related (*E*)-9-(2-nitrovinyl)anthracenes as antiproliferative compounds in CLL, which is a more common but related B-cell malignancy. Nitro-group-containing compounds may induce selective cancer cell toxicity by diverse mechanisms [46] such as topoisomerase inhibition [47], histone deacetylase inhibition [48], DNA alkylation [49] or tubulin polymerisation inhibition [50,51,52], while anthracene-containing compounds are reported to interact selectively with G-quadruplex structures and inhibit telomerase [53].

The objective of this research was the investigation of a series of (*E*)-9-(2-nitrovinyl)anthracenes and related nitrostyrene compounds for antiprolifertive evaluation in BL and CLL cell lines and is arranged as follows:i.The synthesis of a panel of nitrovinylanthracenes for initial BL evaluation to optimise the core structure for further CLL investigation;ii.The identification of a focussed panel of nitrostyrenes to confirm the efficacy of nitrostyrene pharmacophore in CLL;iii.The evaluation and optimisation of the antiproliferative activity of the selected nitrovinylanthracenes in CLL and related cell lines;iv.The determination of the pro-apoptotic effects of nitrovinylanthracenes in CLL cells.

In this work, we initially prepared a series of novel halogenated nitrostyrenes for evaluation in BL to confirm the requirement of the nitrostyrene pharmacophore for antiproliferative activity in CLL, based on our previous investigations [39]. Subsequently, a series of substituted 9-anthraldehydes were synthesised from the appropriate anthracenes, anthrones and anthraquinones including chloro, bromo, methyl, phenyl, methoxy and isopropyl substitutions at C-10, from which the required (*E*)-9-(2-nitrovinyl)anthracenes were synthesised. The reduction of (*E*)-9-(2-nitrovinyl)anthracene was also investigated together with inclusion of the phenanthrene system and the extension of carbon chain between the nitrovinyl unit and the anthracene ring. Other unsaturated systems introduced at C-9 include oximes, cyanovinyl and nitrone systems to assess their effect on the activity of the series. The target substituted anthracene structures identified for investigation are summarised in Figure 3.

## 2. Chemistry

The (*E*)-(2-nitroprop-1-en-1-yl)benzenes and (*E*)-(2-nitrobut-1-en-1-yl)benzenes (**11a–l**) were prepared by the microwave assisted Henry–Knoevenagel condensation reaction of nitroethane or nitropropane with the appropriately chloro- and fluoro-substituted aryl aldehydes catalysed by cyclohexylamine (31–65%) as reported in [39] (Figure 1). Nitrostyrenes in common with many alkene systems can exist as either *cis* (*Z*) or *trans* (*E*) isomers. However, in these Henry–Knoevenagel condensation reactions, the *trans* (*E*) configuration is sterically favoured, and the related *cis* (*Z*) isomer was not isolated in these products [39].

A panel of diversely substituted (*E*)-9-(2-nitrovinyl)anthracenes and related compounds was next prepared for evaluation. The appropriate substituted-9-anthraldehydes were first synthesised from the corresponding anthracenes, anthrones and anthraquinones (Figure 2) [54]. The Grignard reaction of alkyl or aryl magnesium bromides with anthrone **12** was exploited to prepare the required substituted anthracenes [55] and gave 9-substituted anthracenes **13a–c** in excellent yields (step *a*, 76–84%). These substituted anthracenes were subsequently formylated using the Vilsmeier–Haack reagent to give 10-substituted-9-anthraldehydes **14a–c** in yields of 70–80%, including the novel **14b** (step *b*, Figure 2). 9-Methoxyanthracene **13d** was synthesised from anthrone in an acid-catalysed one-pot methylation reaction with trimethylorthoformate [56] (step *c*, 76% yield), and subsequently formylated to give **14d** as above. The 9-methylanthracene analogue **13e** was prepared from 9-anthraldehyde using a Wolff-Kishner reaction of 9-anthraldehyde with hydrazine and potassium hydroxide, (step *f* 84% yield). Vilsmeier-Haack reaction of **13e** gave 10-substituted-9-anthraldehyde **14e** (step *b*, Figure 2). Other formylation systems previously used for the formylation of anthracene derivatives include the substitution of halides using *N*-methylformanilide/*n*-butyllithium [57] and nucleophilic attack of the imine intermediate produced by aluminum trichloride/tin (IV) chloride [58].

10-Bromoanthracene-9-carbaldehyde **14f** was prepared via a two-step synthetic route from anthracene; 9,10-dibromination of anthracene with bromine in dichloromethane afforded **15** (step *g*) [59]. A subsequent lithium–halogen exchange/formylation reaction with *n*-butyllithium and N-methylformanilide [57] afforded 10-bromoanthracene-9-carbaldehyde **14f** in a 72% yield (step *h*, Figure 2). 9-Anthraldehyde can also be selectively brominated with *N*-bromosuccinimide [60]. 4,5-Dichloroanthracene-9-carbaldehyde **14g** was prepared via a two-step synthetic route, which required the reduction of 1,8-dichloroanthracene-9,10-dione to 1,8-dichloroanthracene **16** with a zinc/ammonia system (step *i*) followed by acidification of the reaction product in a 30% yield. Formylation of **16** gave the desired 4,5-dichloroanthracene-9-carbaldehyde **14g** (step *j*, Figure 2). 2-(Anthracen-9-yl)acetaldehyde **18** was prepared as an *E*/*Z* mixture in two steps from 9-anthraldehyde. A Wittig reaction with (methoxymethyl)triphenylphosphonium chloride, yielded 60% of the enol ether 9-(2-methoxyvinyl)anthracene **17** (step *d*). Cleavage of **17** afforded the 2-(anthracen-9-yl)acetaldehyde **18** in a 83% yield (step *e*, Figure 2) [61].

A series of diversely substituted (*E*)-9-(2-nitrovinyl)anthracenes were prepared from the substituted-9-anthraldehyde library described above, together with some related commercially available anthracene aldehydes (Figure 3). A piperidine-catalysed Henry–Knoevenagel condensation reaction was utilised to obtain the panel of (*E*)-9-(2-nitrovinyl)anthracenes **19a–m** (piperidine acetate, excess nitroalkane, 1.5 h 90 °C) with a significant increase in the product yield (up to 99%) when compared with the alternative method (cyclohexylamine, acetic acid, excess nitroalkane, 20 min, MW) [62] and was particularly useful for preparation of **19c.** The yields for compounds **19a, 19b** and **19c** for the piperidine-catalysed method were 99%, 73% and 60% compared with 25%, 10% and trace, respectively, for the microwave method. The (*E*)-9-(2-nitrovinyl)anthracenes **19a–m** included a range of C-9 and C-10 halogen, alkyl and aryl substitutions and the previously unreported compounds **19h–m**. In the ^1^H NMR spectrum of **19f**, the downfield singlet occurring at δ 8.49 ppm was characteristic of H-1′, the proton on C-1′ of the nitrovinyl group. In the ^13^C NMR spectrum, the signal at 130.1 ppm was allocated to the nitrovinyl C-1′, while the quaternary signal at 157.0 ppm was assigned to C-2′ due to the adjacent electron-withdrawing nitro group (Appendix A).

Alkyl substituents (methyl and ethyl) were introduced at C-2′ by using nitroethane and nitrobutane, respectively, in the Henry–Knoevenagel condensation reaction to afford **19b, 19c, 19e** and **19f,** while the reaction of the 2-(anthracen-9-yl)acetaldehyde with nitromethane and nitroethane allowed the introduction of an additional carbon linker in novel compounds **20a** and **20b** (Figure 3). In the ^1^H NMR spectrum of (*E*)-9-(3-nitroallyl)anthracene **20a**, the C-1′ protons were identified as a doublet at δ 4.58 (*J =* 6.1 Hz), the nitrovinyl H-3′ at δ 6.62 (dd, *J =* 13.4, 1.8 Hz), the large *J* value indicating *trans* coupling for the nitrovinyl protons and the alkene H-2′ as a multiplet δ 7.66–7.77. In the ^13^C NMR spectrum of **20a**, the C-1′ methylene signal at 27.0 ppm was inverted in the DEPT 135 NMR spectrum. A C-H COSY experiment confirmed the correlation of the CH resonances at 140.7 ppm and 141.1 ppm to the multiplet at δ 7.66–7.77 and the double doublet at δ 6.62 in the ^1^H NMR spectrum (see Appendix A). The signals at 140.7 ppm and 141.1 ppm were assigned to the nitrovinyl C-2′ and C-3′, respectively. (*E*)-9-(2-Nitrovinyl)phenanthrene **21** was obtained by the Henry–Knoevenagel condensation of 9-phenanthraldehyde and nitromethane in a 15% yield [63], while reduction of the nitrovinyl compound **19a** with sodium borohydride afforded the novel nitroethylanthracene **22**, (Figure 3).

The X-ray structure of 9-chloro-10-(2-nitrobut-1-en-1-yl)anthracene **19f** was determined (Figure 4); in the asymmetric unit, the (*E*)-configuration of the nitrovinyl unit was observed. Figure 4B represents the molecular structure of compound **19f** and confirmed the (*E*)-configuration of the nitrovinyl group together with the π-π stacking interactions (Figure 4C). The bond distances and torsional angles determined for the aromatic ring and the nitrobutene substituents in compound **19f** were in close agreement with the reported values for related β-methyl-β-nitrostyrenes [39,64,65]. The nitrostyrene C=C bond was observed at 1.323(3)Å for compound **19f** (C(15)-C(16)), similar to 1.332(2) Å reported for the (*E*)-1-(3,4-methylene-dioxy-6-fluorophenyl)-2-nitropropene [64]. The nitro and ethyl substituents on the alkene were out of the plane of the anthracene ring, with a torsional angle for C(7)-(C8)-(C15)-(C16) of 108.8(3)° (Figure 4 and Appendix A). The crystal and experimental data for compound **19f** are shown in Table 1.

The 3-(anthracen-9-yl)acrylonitrile derivatives **23a–c** were obtained in 40–95% yields, respectively, on reaction of the 9-anthraldehydes with activated methylenecyano compounds via the base-catalysed Knoevenagel condensation reaction (step *a* and *b*, Figure 4). In the ^1^H NMR spectrum of novel compound **23b,** the downfield singlet at δ 8.91 was assigned to the cyanovinyl H-1′. The ^13^C NMR spectrum confirmed quaternary signals at 93.4 ppm, 111.1 ppm and 112.7 ppm for the cyanovinyl carbons C-2′, C-1″ and C-2″; the downfield signal at 160.2 ppm was assigned to the cyanovinyl C-1′. In further reactions, condensation of 9-anthraldehyde and acetone afforded 4-(anthracen-9-yl)but-3-en-2-one **24** (step *c*) [66], while **25** was isolated upon reaction of 9-anthraldehyde with Meldrum’s acid (step *d*) [67] (Figure 4). The nitrile **28** [68] was obtained by reduction of 9-anthraldehyde to the anthracen-9-ylmethanol **26** (step *e*) [69], followed by chlorination (step *f*) and subsequent treatment of the chloride **27** with potassium cyanide (step *g*). The (anthracen-9-yl)methylamines **29a–d** were prepared by nucleophilic substitution of 9-(chloromethyl)anthracene **27** with selected amines with a 75–90% yield (step *h*, Figure 4).

The anthracene-based 4-(anthracen-9-yl)-1*H*-1,2,3-triazoles **30a** and novel **30b** were obtained by reaction of the (*E*)-9-(2-nitrovinyl)anthracenes **19a** and **19b** with sodium azide in DMSO (Figure 5). However, during attempts to optimise this reaction for **30a**, an unexpected novel product was isolated: 4-(anthracen-9-yl)-4,5-dihydro-1*H*-1,2,3-triazole **31** in a 90% yield. This triazole is a reduced version of the expected product **30a**. It is possible that increasing the reaction time from 1 to 12 h allowed for a slower reduction reaction to occur in the presence of DMSO and sodium azide. Reductive conditions of hot DMSO and sodium azide solutions have been previously reported with the sodium metal required for the triazole reduction provided by the azide salt [70]. 4-(Anthracen-9-yl)-1*H*-1,2,3-triazoles have been synthesised from 9-ethynlyanthracene, trimethylsilyl azide and copper iodide [71], and also by ZrCl_4_-mediated conversion of vinyl nitrates to 1,2,3-triazoles [72]. In the ^1^H NMR spectrum of **31,** the doublet (δ 4.64, *J =* 4.9 Hz) was assigned to the H-5′ protons, the triplet (δ 6.68, *J =* 5.2 Hz) was assigned to the H-4′ and the downfield singlet δ 11.40 corresponded to the triazole NH.

X-ray crystallography confirmed the structure of 4-(anthracen-9-yl)-1*H*-1,2,3-triazole **30a**, (Figure 5 and Appendix A). The triazole bond lengths and angles were within the reported ranges [73]. The 1,2,3-triazole N=N bond N(2)-N(3) was observed at 1.31(4) Å, N(1)-C(5) at 1.345(11) Å, N(1)-N(2) at 1.354(16) Å, N(3)-C(4) at 1.373(5) Å and the C(4)-C(5) C=C bond at 1.374(11) Å. The C6-C4 bond linking the triazole to the anthracene was 1.475(6) Å, with a torsional angle of −110.7°. The triazole ring N1-N2-N3-C4-C5 made a dihedral angle of ca. 110.8° with the anthracene ring C6-C19. The crystal and experimental data for compound **30a** are shown in Table 1.

To assess the antiproliferative effect of alternative isosteric imine systems to replace the nitrostyrene, the anthracen-9-ylmethanimines **32a–d** were prepared by reaction of 9-anthraldehyde, 9-chloroanthraldehyde or 2-(anthracen-9-yl)acetaldehyde **18** with the appropriate amine in yields of 85–90%, (Figure 5). A series of anthracene amides **33a–f** were obtained by coupling anthracene-9-carboxylic acid and selected amines using the Mukaiyama reaction (75–90%), as shown in Figure 5. In the ^1^H NMR spectrum of the novel anthracen-9-yl(azepan-1-yl)methanone **33c**, a triplet at δ 3.09 (*J =* 6.1 Hz) and a multiplet (δ 4.02) were assigned to the C_1_′ and C_6_′ methylene protons due to the non-equivalent nature of these homopiperidine protons, with signals at 45.1 ppm and 49.0 ppm in the ^13^C NMR spectrum assigned to C_1_′ and C_6_′ (see Appendix A).

In total, a panel of 58 anthracene-based compounds and 12 halo-substituted nitrostyrenes were designed and synthesised for this study. Reaction of a number of substituted anthraldehydes with the required nitroalkanes in a Henry–Knoevenagel condensation reaction afforded a series of 16 substituted (*E*)-9-(2-nitrovinyl)anthracenes and phenanthrenes. Anthracene ring substitutions included methoxy, isopropy, alkyl (methyl, ethyl), phenyl and halogen (chlorine and bromine). The alkyl substituent at C-2 was varied by using different nitroalkanes (nitromethane, nitroethane and nitropropane). Extension of the carbon linker between the vinyl unit and the anthracene moiety was also achieved using a Wittig reaction. Related vinyl functionalities were introduced on the anthracene including cyanovinyl, oxime, hydrazone and alkyl analogues, together with a selection of amines and amides. The compounds synthesised were initially screened for biological activity in the BL cell lines DG-75 and MUTU-I with subsequent screening of the more potent compounds in the CLL cell lines PGA-1 and HG-3. The results of this preliminary screen are discussed in the following section.

### 2.1. In Vitro Antiproliferative Activity of Nitrosytrenes, Nitrovinylanthracenes and Related Compounds in Burkitt Lymphoma

A panel of 58 anthracene-based compounds were initially evaluated for anti-proliferative activity in the Burkitt lymphoma EBV^−^ MUTU-I (chemosensitive) and EBV^+^ DG-75 (chemoresistant) cell lines at 10 µM and 1 µM using an alamarBlue viability assay to determine the structure–activity relationships for these anthracene compounds and to identify the most potent compounds for further investigation. We previously reported the anti-proliferative effects of a panel of nitrovinylstyrenes in the BL MUTU-I and DG-75 cell lines and identified halogenated compounds **11c**, **11f–l** with cell viability at 10 µM in the range 2–26% and 0–16%, respectively, and with IC_50_ values in the range 0.82–2.18 µM (MUTU-I) and 2.05–3.11 µM (DG-75) [39] (Appendix A), suggesting that the nitrovinylstyrene pharmacophore may be suitable for further study. In this work, the antiproliferative activity of the more potent selected nitrostyrene compounds **11i, 11h, 11g** and **11j** was further investigated in the following BL cell lines: Ramos (BL, EBV-negative) and Bjab (BL, EBV-negative) together with HeLa (cervical), MCF-7 (ER-positive breast cancer) and HL-60 (promyelocytic leukaemia) cell lines (Table 2). The compounds elicited good anti-proliferative effects at a 10 µM concentration in all cell lines, e.g., 4.4–13.9% viability in the BL Ramos and 6.5–16.55% in the leukaemia HL60 cell line, exerting a more potent effect than taxol in all of these cell lines, apart from MUTU-I. (*E*)-1-Chloro-2-(2-nitrobut-1-en-1-yl)benzene **11j** was particularly potent at 10 µM in the Ramos BL cell line (4.4% cell viability) and 9.1% viability in the HL60 cell line.

### 2.2. Effect of Nitrostyrene ***11h*** on the Viability of PBMCs

The nitrostyrene **11h** was evaluated for its cytotoxic effect on healthy donor peripheral blood mononuclear cells (PBMCs) to determine the selective toxicity of compounds containing the nitrostyrene pharmacophore on malignant BL cell lines over normal blood cells. Compound **11h** was evaluated at 1 μM and 10 μM concentrations over a 24 h treatment time (Appendix A). Compound **11h** demonstrated a low toxicity in PBMCs at 1 μM (74% viable cells remaining). In comparison, compound **11h** induced a significant anti-proliferative effect in MUTU-I cells, with 39.8% viable cells remaining at 1 μM. A similar response was observed in DG-75 cells at the higher concentration (10 μM); a potent anti-proliferative effect (0.038% viable cells remaining) was observed, in comparison to 34.1% of viable PBMCs, indicating that compound **11h** is selectively toxic to these BL cell lines.

### 2.3. Effect of Pre-Treatment with N-Acetylcysteine and Caspase Inhibitor Z-VAD-FMK on Induction of Apoptosis by Compound ***11h***

Additional annexin V/PI FACS analysis was carried out in the presence of a reactive oxygen species (ROS) inhibitor (N-acetylcysteine) and a pan-caspase inhibitor (Z-VAD-FMK) in order to study their effects (if any) on the pro-apoptotic effects of **11h** (Figure 6). In the presence of NAC (5 μM), the apoptosis induced by compound **11h** decreased from 68% to 21% at 2.5 µM and from 88% to 33% at 5 µM in the PGA1 cell line. Similar results were obtained in the HG-3 cell line at 2.5 µM (90% to 31%) and 5 µM of **11h** (84% to 51%). In the presence of the caspase inhibitor Z-VAD-FMK (5 μM), the apoptosis induced by compound **11h** decreased from 63% to 25% at 2.5 µM and from 79% to 43% at 5 µM in the PGA1 cell line. Similar results were obtained in the HG-3 cell line at 2.5 µM (84% to 29%) and 5 µM (74% to 50%). These findings indicate that both caspases and ROS may be involved in the mechanism of apoptosis for compound **11h**.

### 2.4. In Vitro Antiproliferative Activity of the Nitrovinylanthracenes and Related Compounds in Burkitt’s Lymphoma

As a further development, the anticancer effects of the panel of nitrovinylanthracene and related compounds synthesised together with the C-9 substituted anthracenes such as amines, carboxamides, cyanovinyl and hydrazone derivatives were investigated. The effects of additional C-10 substitution (e.g., alkyl, alkoxy, halogen, aryl) on the anti-proliferative effects of the anthracene compounds were also evaluated and are discussed by structural type. The results obtained from this preliminary screen in the MUTU-I and DG-75 cell lines (at 10 μM and 1 μM) are displayed in Figure 6 and Figure 7, with maprotiline and taxol used as the positive controls. Maprotiline induced a modest anti-proliferative effect at 10 μM in the MUTU-I and DG-75 BL cell lines (72% and 65% viable cells, respectively). Treatment with taxol resulted in a 7% cell viability at 10 μM and 32% at 1 μM in MUTU-I cell line, while a 40% and >90% cell viability was observed at 10 and 1 μM treatment concentrations, respectively, in the more chemoresistent DG-75 cells. The lead nitrovinylanthracene compound **19a** (IC_50_ 2.57 µM in MUTU-I and 2.08 µM in DG-75) was more potent than maprotiline (IC_50_ values of 15.8 µM (MUTU-I) and 37.5 µM (DG-75)) in both BL cell lines and compared favourably with taxol in these cell lines (IC_50_ 0.32 µM in MUTU-I and 1.32 µM in DG-75) [41] (Figure 7A,B). The 10-substituted (*E*)-9-(2-nitrovinyl)anthracenes **19g–m** were found to elicit similar potent anti-proliferative effects to lead compound **19a** in both the MUTU-I and DG-75 BL cell lines at both concentrations, with <20% cells remaining. This compared well to maprotiline (>60% viable cells remaining in both cell lines at 10 µM) and resulted in the identification of a potent series of active compounds for further investigation. Extension of the alkyl chain (methyl, ethyl) at C-2 of the nitrovinyl unit deactivated the (*E*)-9-(2-nitrovinyl)anthracene pharmacophore (>80% viability), and this trend was observed in compounds (**19a**, **19b**, **19c**) and (**19d, 19e, 19f**). Introduction of C-10 chloro (compound **19d**) increased the inhibition of MUTU-I cell growth at 1 µM when compared to **19a** (<20%) but reduced the anti-proliferative effect in the DG-75 cell line, an effect also observed for the C-9 bromo compound **19h** (Figure 7A,B). Introduction of the phenanthrene moiety **21** reduced the anticancer effect in the DG-75 cell line to 60% cell viability. The extended-chain nitrovinyl analogues **20a** and **20b** induced good activity at 10 µM in the MUTU-I cell line (~15–40%); however, a reduction in activity was observed in the DG-75 cell line (79–97% viability). Reduction of the vinyl unit of compound **19a** to give compound **22** was detrimental to the anticancer effects with activity lost in both BL cell lines (>85%) (Figure 7A,B). It was concluded that the nitrovinyl unit was required for optimal anti-proliferative activity.

The (*E*)-3-(anthracen-9-yl)acrylonitriles **23a** and **c** and 2-(anthracen-9-yl)acetonitrile **28** (Figure 7A,B), (anthracen-9-yl)methylamines **29a–d** and their precursors **26** and **27** (Figure 8A,B), anthracen-9-ylmethanimines **32a–e** (including hydrazones, oximes and nitrones) (Figure 8A,B) and anthracene amides **33a–f** (Figure 8A,B) exhibited poor anti-proliferative effects in the MUTU-I and DG-75 cell lines at 10 µM (>80% and >60% viability, respectively), indicating the requirement of the nitrovinyl functionality for activity. The 4-(anthracen-9-yl)-1*H*-1,2,3-triazoles **30a, 30b** and **31** with a constrained (*E*)-configuration for the vinyl system and compounds **24** and **25** with alternative vinyl functionalities were inactive in both the BL cell lines (>90% viability) (Figure 7A,B). However, the C-9 dicyanovinyl compound **23b** (synthesised to evaluate the effects of alternative substituents at the C-2 carbon of the vinyl unit) was effective at 10 µM in the MUTU-I cell line (~20% viability) with moderate activity at 10 µM in the DG-75 cell line (~55%) Figure 7A,B). (See Appendix A for complete cell viability data for all compounds).

### 2.5. Physicochemical, ADME, Pharmacokinetic and Stability Properties of (E)-9-(2-Nitrovinyl)Anthracenes and Related Compounds

The physicochemical, ADME and pharmacokinetic properties of the most potent synthesised compounds (**19a–19m, 20b, 21, 22, 23a, 23b**) were initially investigated using Pipeline Pilot Professional [74] (see Appendix A for details of the Tier 1 profiling screen, Appendix A). These anthracene compounds complied with Lipinski and Veber rules with a molecular weight less of than 500 Da (within the range of 249–328 Da) and with fewer than 10 rotatable bonds, fewer than 10 hydrogen bond acceptors, fewer than 5 hydrogen bond donors and a logP of less than 5 (in the range 2.68–4.00) (Appendix A). The topological polar surface area (TPSA) of the compounds was found in the range 45.82–47.58 Å^2^, within the desirable limit of 140 Å^2^ for good intestinal absorption. The compounds were predicted not to inhibit CYP2D6, while high blood–brain barrier (BBB) absorption levels and good plasma protein binding properties (greater than 90%) were predicted for all compounds (Appendix A). The synthesised compounds **19a–19m, 20b, 21, 23a** and **23b** are predicted to be un-ionised at physiological pH, with the theoretical pKaH value for compound **22** calculated with a Marvin of 8.22. However, low aqueous solubility is predicted for the panel of compounds in the range logSw = −7.0840 to −5.3960, e.g., the 10-methoxy compound **19g** is predicted with greatest solubility in the series (logSw = −5.3960 mol/L) (see Appendix A). The nitrovinylanthracene compounds **19a–19m, 20b, 21, 22, 23a** and **23b** were not signalled in a filter for pan-assay interference compounds (PAINS) [75] and are predicted to have good drug-like physicochemical properties within the appropriate range for oral bioavailability [76,77]. Additional biochemical studies are described in the following sections to determine their mechanism of action.

A preliminary HPLC stability study was performed on a representative nitrovinylanthracene compound **19m** (isopropyl) in various biologically relevant pH systems (acidic pH 4 found in the stomach, basic pH 9 found in the intestine and pH 7.4 in the plasma). The half-life (t_1⁄2_) was determined to be 19 h at pH 9 (42% remaining at 24 h) and greater than 24 h at both pH 4 and pH 7.4, with 55% and 56% remaining, respectively. Based on the results of this stability study, the compound **19m** was determined to be suitable for further preclinical investigation.

### 2.6. Evaluation of In Vitro Antiproliferative Activity of Nitrostyrenes and Anthracene-Based Maprotiline Analogues in Chronic Lymphocytic Leukaemia (CLL)

The panel of nitrostyrenes and anthracene-based maprotiline analogues was next evaluated for in vitro anti-proliferative activity in CLL. The HG-3 cell line was established from an in vitro EBV (Epstein–Barr virus) infection from an IGVH1–2 unmutated B1 lymphocyte origin CLL patient clone and is representative of poor patient prognosis [78]. The PGA-1 cell line is a cell line that was established from leukemic B cells of a Caucasian male with CLL with a mutated IGVH1-2 and is representative of good patient prognosis [79]. Fludarabine phosphate was used as a comparative control for CLL cell lines [45] (IC_50_ values of 28.1 μM and 32.0 μM in HG-3 and PGA-1 cell lines, respectively, with cell viability of HG-3 60%, PGA-1 65% at 10 μM concentration). From our previous work, we have identified a number of nitrostyrene-containing compounds demonstrating antiproliferative activity in BL cells, e.g., compounds (**10a–c**), with IC_50_ values of 0.45, 0.47 and 2.97 μM in MUTU-I and IC_50_ values of 1.41, 1.92 and 6.39 μM in DG-75, respectively, while anthracene (**19a**) also demonstrated activity in BL cell lines with IC_50_ values of 3.0 μM (MUTU) and 1.5 μM (DG75) [41], suggesting that the nitrostyrene pharmacophore is relevant in the antiproliferative activity of the series.

### 2.7. Antiproliferative Activity of Nitrostyrenes in HG-3 and PGA-1 CLL Cell Lines

The halogenated nitrostyrenes **11a–l** were initially screened in CLL cells PGA-1 and HG-3 at 1 and 10 μM together with lead nitrostyrene compound **10a** from our previous study in BL. All compounds displayed low PGA1 and HG3 viability at 10 μM, while the control drug fludarabine **1** demonstrated 89% (HG-3) and 94% (PGA-1) viability at 10 μM. The (*E*)-(2-nitrovinyl)benzenes) **11a–l** confirmed the nitrostyrene moiety as a promising pharmacophore for CLL activity and facilitated a comparison of the effects of halogen (*ortho*, *para* and *meta* Cl or F) and alkyl substituents (C-1 methyl or ethyl) on cell viability. All compounds displayed good antiproliferative activity in the HG-3 cell line at 10 μM with 0.11–11.15% viability (Figure 9A) and also in PGA-1 (0.018–7.05% viability) (Figure 9B), with the fluoro-substituted compounds being more active than the corresponding chloro compounds. For the fluoro-substituted fluoro-(2-nitrobut-1-en-1-yl)benzenes **11d–f**, the *ortho*-substituted **11d** showed the greatest antiproliferative activity (0.24% cell viability at 10 μM and 14.5% at 1 μM). The *meta*-substituted **11k** (*E*)-2-chloro-(2-nitrobut-1-en-1-yl)benzene was identified as the most potent of the chloro series in the HG-3 cell line with cell viabilities of 0.34% (10 μM) and 60.7% (1 μM) (Figure 9A). Cell viability values in the PGA-1 cell line were approximately 20% lower than those observed in the HG-3 cells, with the chloro-substituted (*E*)-(2-nitrovinyl)benzenes displaying greater potency than the fluoro-substituted series in the PGA-1 cell line. The (2-nitrobut-1-en-1-yl)benzene compounds **11j** and **11k** were identified as the most potent in the series (0.43% and 0.15% cell viability at 10 μM and 6.31% and 3.26% at 1 μM, respectively), showing superior activity when compared with the lead compound **10a** with a cell viability of 96.4% (1 μM) and 0.97% (10 μM), see Figure 9B.

### 2.8. Antiproliferative Activity of Anthracene-Based Maprotiline Analogues in HG-3 CLL Cell Line

The most potent (*E*)-9-(2-nitrovinyl)anthracenes identified from the BL screen (**19a–j**, **19l**, **19m**, **20b**, **21**, **22, 23a, 23b**) were next evaluated in CLL HG-3 cells at 1 and 10 μM concentrations (Figure 10). Compounds **19a**, **19g 19i**, **19l** and **19m** were very effective in the HG-3 cells, with a 0.5–8.3% viability demonstrated at a 10 µM treatment concentration (Figure 10A). The 10-ethyl (**19l**) and 10-isopropylanthracene (**19m**) compounds were identified as the most potent compounds, with a 0.5% and 0.9% viability at 10 µM and 78% and 82% viability at 1 μM, respectively (Figure 10A). Reduction of the nitrovinyl group of **19a** to the nitroethane compound **22** resulted in a fifteen-fold significant loss in potency (76% cell viability). The remaining (*E*)-9-(2-nitrovinyl)anthracene derivatives with aryl substituents at C-10 were significantly less potent in the HG-3 cell line and are shown in Figure 10A. At the 10 µM treatment concentration, the most effective compound of this remaining group was identified as the phenanthrene derivative **21** (29% viability), while the 10-chloro compounds **19d** (10-chloro-2-nitrovinyl derivative) and **19e** (10-chloro-3-propenyl derivative) demonstrated very poor activity (97% and 87% viability, respectively). The introduction of an alkyl substituent on the nitrovinyl group generally resulted in a significant reduction in activity for the compounds, e.g., comparing the activity of extension of the alkyl chain on the 2-nitrovinyl unit resulted in a dramatic decrease in activity (compounds **19a**, **19b** and **19c** with 8%, 82% and 72% cell viability, respectively, at 10 µM). A similar trend was observed with compounds **19d**, **19e** and **19f**. Introduction of the 10-chloro substitution on **19d** resulted in a loss in activity (97% cell viability) when compared to the unsubstituted analogue **19a** (cell viability 8%).

### 2.9. Antiproliferative Activity of Anthracene-Based Maprotiline Analogues in PGA-1 CLL Cell Line

The cell viability results of the (*E*)-9-(2-nitrovinyl)anthracenes in the PGA-1 cell line are shown in Figure 10B. At the 10 µM treatment concentration, all compounds **19a**, **19g, 19i**, **19l**, **19m** and **21** were very effective, with **19g** (10-methoxy 1.21%), **19l** (10-ethyl, 2.7%) **19m** (10-isopropyl, 2.9%) and **19i** (10-methyl, 2.9%) being identified as the most potent with the unsubstituted compound **19a** (5.1%) and the phenanthrene derivative **21** demonstrating an 11.8% cell viability, while compounds **19h** (10-bromo) and **19d** (9-chloro) demonstrated reduced activity with cell viability values of 36.9% and 51.8%, respectively. The remaining compounds in the series, including the reduced **22** (75% viability), 9-(3-nitroallyl)anthracene **20a** and acrylonitriles **22a** and **22b,** demonstrated poor activity with viability >50%. These results indicate the requirement for the intact double bond of the nitrostyrene for the antiproliferative activity. At the 1 µM treatment concentration, the 10-methoxy **19g** compound was also the most active (43% viability). As observed in the HG-3 cell line, introduction of a methyl or ethyl substituent on the nitrovinyl group resulted in significant decrease in activity (compounds **19a**, **19b** and **19c** with 5.1%, 81% and 66% cell viability values, respectively, at 10 µM), with a similar effect for compounds **19d**, **19e** and **19f**. Introduction of the ring Cl at C-10 of the anthracene series, e.g., compounds **19d–f**, resulted in a significant decrease in activity in both cell lines (cell viability in the range 86–100% at 10 μM in HG-3 and 51–100% at 10 μM in PGA-1 cells) when compared with the chloro-substituted nitrostyrenes **11g–l** (cell viability 0.1–6% at 10 μM in HG-3 and 0.1–0.5% at 10 μM in PGA-1 cells).

The most potent compounds from the series **19a, 19g, 19i, 19l** and **19m** were chosen for IC_50_ determination and evaluation in the CLL cell lines PGA-1 and HG-3.

### 2.10. In Vitro IC_50_ Determination of the Most Potent (E)-9-(2-nitrovinyl)anthracene Derivatives in HG-3 Cells and PGA-1 Cells

The IC_50_ values at 24 h for the selected (*E*)-9-(2-nitrovinyl)anthracene compounds **19a, 19g, 19i, 19l** and **19m** in HG-3 cells and PGA-1 cells were determined using a concentration range of 10 μM–0.01 μM (Table 3). The compounds demonstrated a more potent effect than the fludarabine control (5–40 fold greater in the HG-3 cells; 4–25 fold greater in the PGA-1 cells) across both the HG-3 and PGA-1 cell lines with IC_50_ ranges of 0.70–3.85 μM and 1.29–9.10 μM, respectively. In the HG-3 cells, the most potent compounds identified were the 10-methoxy derivative **19g** (IC_50_ 0.17 μM) and the 10-isopropyl derivative **19m** (IC_50_ 0.70 μM), while in the PGA-1 cells, the most potent compounds were **19g** (IC_50_ 1.29 μM) and the 10-ethyl derivative **19l** (IC_50_ 1.30 μM). These results suggest that alkyl and alkoxy substituents present at the C-10 position on the anthracene core can lead to a greater anti-proliferative activity in CLL compared to the unsubstituted **19a**. The 10-isopropyl substituent of **19m** was selective, causing a 13-fold greater response in the HG-3 as opposed to PGA-1 cells (0.7 μM vs. 9.1 μM, possibly due to increased lipophilic/steric effects) and approximately a 3.5-fold increased activity compared to **19a** in both cell lines. Furthermore, **19g** (10-methoxy) resulted in a 7.8-fold improvement in the IC_50_ value in the HG-3 compared to PGA-1 cells and a similar activity to **19l** (10-ethyl) in the PGA-1 cells (IC_50_ = 1.29 μM). These results suggest the potential for similar, yet distinct, compound attributes for potent antiproliferative activity in the main two CLL disease cell subtypes.

### 2.11. In Vitro Antiproliferative Activity of Nitrovinylanthracenes in Estrogen-Receptor-Positive Breast Cancer Cell Lines MCF-7 and MDA-MB-231

Selected compounds were also evaluated in the estrogen-receptor-positive (ER+) breast cancer cell line MCF-7 and the triple-negative breast cancer (TNBC) cell line MDA-MB-231. TNBC accounts for 10–15% of breast cancers that do not express estrogen and progesterone receptors (ER/PR) and are HER2-negative. TNBCs are not responsive to hormone therapies, e.g., the selective estrogen receptor modulator tamoxifen, the aromatase inhibitor anastrozole or the monoclonal antibody herceptin, which targets the HER2 receptor (human epidermal growth factor receptor 2). Fewer treatment options are available for TNBC compared with ER+, PR+ and HER2+ breast cancers, and the outcome is uncertain [80]. Five compounds were screened in MCF-7 and MDA-MB-231 breast cancer cells at 1 and 10 μM concentrations (Figure 11), and based on the results, the IC_50_ values for three of the five compounds were determined (Table 4). The nitrovinylanthracenes **19a**, **19g** and **19i** were found to display moderate antiproliferative activity in MCF-7 breast cancer cells with an IC_50_ value of 1.85 μM for the most potent example **19a**, which compared favourably with tamoxifen (IC_50_ = 4.12 μM). The compounds also displayed a low micromolar activity when evaluated in the TNBC cell line MDA-MB-231 with IC_50_ values in the range 3.26–3.82 μM, suggesting that they were not selective for ER+ breast cancer cells.

### 2.12. Pro-Apoptotic Effects of Nitrostyrene Compounds ***11g***, ***11h***, ***11i***, ***11j***, ***11k*** and ***11l*** in MUTU-I and DG-75 BL Cell Lines

To examine the potential anti-proliferative effects of these nitrostyrene-type compounds, the ability of the most potent compounds identified from the cell viability study to induce apoptosis in the MUTU-I and DG-75 cell lines was investigated using Annexin V and propidium iodide staining at a 10 μM concentration. The chemotherapeutic drug taxol was used as a positive control. The pro-apoptotic effects of the selected nitrostyrene compounds **11g, 11h, 11i, 11j**, **11k** and **11l** (10 μM) were determined using FITC (fluorescein isothiocyanate), Annexin V/PI (propidium iodide) staining and FACS (fluorescence-activated cell sorting) analysis to characterise the mode of cellular death induced by the synthesised compounds. Four populations were produced in this assay: Annexin-V- and PI-negative (Q4, healthy cells), Annexin-V-positive and PI-negative (Q3, early apoptosis), Annexin-V- and PI-positive (Q2, late apoptosis) and Annexin-V-negative and PI-positive (Q1, necrosis), which were easily identified and quantified. Apoptosis was assessed as % total apoptosis by a combination of early and late apoptosis (Q3 and Q2, respectively). In the MUTU-I cell line, the selected nitrostyrene compounds **11g, 11h, 11i, 11j**, **11k** and **11l** demonstrated a significant increase in apoptosis (80–91%), with the most potent effect being observed for the 3-chloro compound **11k** (91%) and was comparable to the effect induced by taxol (87%) at the same concentration (Table 5). In the DG-75 cell line, the 3-chloro compound **11h** (10 μM) produced a significant increase in apoptosis (92%), while a significant increase in apoptosis (70–92%) in the DG-75 cells was observed upon treatment with the compounds **11g, 11h, 11i, 11j**, **11k** and **11l** and compared favourablely with taxol (72.7%) at 10 μM. The identification of compounds that can induce apoptosis in cancer cells is required in the development of potential lead structures for anticancer drugs.

### 2.13. Pro-Apoptotic Effects of Nitrovinylanthracene Compounds ***19a***, ***19g***, ***19i***, ***19l*** and ***19m*** in HG-3 and PGA-1 CLL Cell Lines

The pro-apoptotic effects of the selected nitrovinylanthracene compounds **19a, 19g, 19i, 19l** and **19m** after 48 h were next determined using FITC (fluorescein isothiocyanate), Annexin V/PI (propidium iodide) staining and FACS (fluorescence-activated cell sorting) analysis to characterize the mode of cellular death induced. Apoptosis was assessed as above using the cell lines HG-3 and PGA-1 and using nitrovinylanthracene compound treatment concentrations of 1 µM and 10 µM. We previously reported that fludarabine phosphate (50 μM) induces an increase of 24.6% in apoptosis of cancer cells isolated from CLL patients [45].

The results from the Annexin V/PI studies of the lead (*E*)-9-(2-nitrovinyl)anthracene compound **19a** (10 µM, 5 µM and 1 µM) and **19g, 19i**, **19l** and **19m** (10 μM and 1 μM) concentrations are shown in Figure 12. In the HG-3 cells, all the compounds tested produced a marked proapoptotic effect. Compound **19a** produced significant concentration-dependent apoptosis, with 93% apoptosis at a 10 μM concentration, 69% apoptosis at 5 μM and 21% apoptosis at 1 μM. Compound **19l** (10-ethyl) produced a 74% apoptotic response at 10 μM followed by **19i** (10-methyl) and **19m** (10-isopropyl) producing 70% and 69% total apoptosis, respectively. Compound **19g** (10-methoxy) produced the lowest pro-apoptotic response at 10 μM, with 54% total apoptosis. In the PGA-1 cells, all the compounds tested also produced a marked concentration-dependent pro-apoptotic effect: **19a** produced 94% apoptosis (10 μM), 67% apoptosis (5 μM) and 40% apoptosis at a 1 μM concentration. Compounds **19i** (10-methyl) and **19l** (10-isopropyl) demonstrated 80% and 82% total apoptosis, respectively, at 10 μM, followed by **19m** (74%) and **19g** (52%) (Figure 12). In both cell lines, alkyl substitution at position 10 of the anthracene scaffold structure was observed to promote a more favourable pro-apoptotic action in the CLL cell lines. The introduction of a methoxy group at C-10 in **19g** compared to the alkyl substituent of **19l** suggests the potential role of hydrophobic groups to increase biological activity (30% activity decrease in PGA-1; 20% activity decrease in HG-3 cells). At a lower concentration (1 μM), **19m** was the most potent compound in the HG-3 cell line by a 14% margin in total apoptosis induced (25% apoptosis), followed by **19a** (21% apoptosis) and **19g** (11% apoptosis). Furthermore, decreasing the steric size of the hydrophobic alkyl group resulted in decreasing the pro-apoptotic effect (**19m** (isopropyl) > **19l** (ethyl) > **19i** (methyl)) at the lower compound concentration in both cell lines. Examples of the quadrant diagrams generated by compounds **19a** and **19m** at 10 μM and 1 μM in the HG-3 cells are illustrated in Figure 13. The observed results in the CLL cell lines HG-3 and PGA-1 suggest that these compounds act by a pro-apoptotic mechanism that is concentration-dependent.

### 2.14. Molecular Modelling

The designed 9-nitrovinylanthracenes **19a–m** and related compounds are structurally related to maprotiline and may drive their cellular antiproliferative effect through a similar mechanism of action. In order to examine the structural similarity in more detail, all the compounds described in this work were overlaid on maprotiline using two separate but complementary methodologies. MOE flexible alignment [84] was used in our previous work [85] and is based on several similarity terms, such as hydrogen bond donor/acceptor, aromaticity and partial charge. A stochastic search procedure was used to flexibly align and superimpose similar functional groups in each molecule while sampling the full conformational flexibility of each structure. OpenEye fastROCS [86] is a GPU-based 3D shape similarity method that takes a low-energy conformation of maprotiline as the query molecule and aligns to it each conformation of the compounds in this paper by a solid body optimisation process to maximise the volume overlap. Both colour (feature) and shape similarity were measured with a Tanimoto score with a maximum (best) score of 1, and the overall overlay quality was giving by the Tanimoto combo (Tc) score, which is the sum of these two scores with a maximum (best) value of 2. All databases and reference structures are provided in the Appendix A as sdf or mdb files.

The overlay results obtained for all the compounds in this study are inconclusive but generally indicate that the compounds in this paper may have a similar mechanism of action as maprotiline (Appendix A). In the fastROCS study, considering the cell viability of the BL MUTU-1 cell lines treated at 10 μM, only 5 of the top 12 overlaid compounds decreased the cell viability by over 50% (Appendix A). A similar result was obtained in the analysis of the CLL HG-3 cell viability. The MOE flexible alignment also demonstrated a lack of correlation between the lowest (best) scored compounds and those with the most promising cellular data, both for the CLL and BL cell lines (Appendix A).

A selection of the best (lowest)-scored overlaid structures for the most potent anthracene compounds **19a, 19g, 19i, 19l** and **19m** (displayed as green in their respective overlays) with the lead compound maprotiline **9** (pink) is provided in Table 6, together with the antiproliferative activity in CLL cells. Shared molecular features were clearly identified, e.g., the (*E*)-configuration nitrovinyl pharmacophore located at C-9 that overlays with the methylpropylamine-containing substituent of maprotiline, and the aromatic anthracene structure that overlays with the 9,10-dihydroanthracene core structure of maprotiline. A selection of the best-scored overlaid structures for the most potent nitrostyrene compounds **11c, 11d, 11j** and **11k** is provided in Table 7. The nitrostyrenes mapped closely to the cyclic core of maprotiline rather than to the central ring and along the sidechain as for the anthracenes; again, the correlation with cellular activity in CLL cell lines was not conclusive (See Appendix A).

## 3. Experimental Section

Uncorrected melting points were measured on a Gallenkamp apparatus. Infra-red (IR) spectra were recorded on a Perkin Elmer Spectrum FT-IR 100 spectrometer (Waltham, MA, USA). ^1^H, ^13^C and ^19^F nuclear magnetic resonance spectra (NMR) were recorded at 27 °C on a Bruker DPX 400 spectrometer (Bruker UK Limited, Coventry, UK) (400.13 MHz, ^1^H; 100.61 MHz, ^13^C; 376.47 MHz, ^19^F) in either CDCl_3_ (internal standard tetramethylsilane (TMS)) or CD_3_OD or DMSO-*d_6_*. For CDCl_3_, ^1^H-NMR spectra were assigned relative to the TMS peak at 0.00 ppm, and ^13^C-NMR spectra were assigned relative to the middle CDCl_3_ peak at 77.0 ppm. For CD_3_OD, ^1^H and ^13^C-NMR spectra were assigned relative to the center peaks of the CD_3_OD multiplets at 3.30 ppm and 49.00 ppm, respectively. Coupling constants are reported in Hertz. For ^1^H-NMR assignments, chemical shifts are reported as the shift value (number of protons, description of absorption and coupling constant(s), where applicable). Electrospray ionisation mass spectrometry (ESI-MS) was performed in the positive ion mode on a liquid chromatography time-of-flight mass spectrometer (Micromass LCT, Waters Ltd., Manchester, UK). The samples were introduced to the ion source by an LC system (Waters Alliance 2795, Waters Corporation, Milford, MA, USA) in acetonitrile: water (60:40% *v*/*v*) at 200 µL/min. The capillary voltage of the mass spectrometer was at 3 kV. The sample cone (de-clustering) voltage was set at 40 V. For exact mass determination, the instrument was externally calibrated for the mass range from *m/z* 100 to *m/z* 1000. A lock (reference) mass (*m/z* 556.2771) was used. Mass measurement accuracies of <±5 ppm were obtained. R_f_ values are quoted for thin-layer chromatography on silica gel Merck F-254 plates unless otherwise stated. Flash column chromatography was carried out on Merck Kieselgel 60 (particle size 0.040–0.063 mm). Microwave experiments were carried out using a Discover CEM microwave synthesiser on the standard power setting (300 watts) unless otherwise stated. See Appendix A for preparation and characterisation of compounds previously reported: **11a–l, 13a–e**, **14a**, **14c–e**, **14f–g**, **15**, **16**, **18**, **19a–g**, **21, 23c**, **24–28**, **29a–d**, **30a**, **32a–e**, **33a**, **33d–f** [39,40,55,57,58,66,67,71,87,88,89,90,91,92,93,94,95,96,97,98,99,100,101,102,103,104,105,106,107,108,109,110,111,112,113,114,115].

### 3.1. 10-Isopropylanthracene-9-carbaldehyde (***14b***)

*N*-methylformanilide (2.03 g, 15 mmol) was added to a cooled, stirred solution of 9-isopropylanthracene (1.76 g, 8 mmol) in phosphorus oxychloride (2.30 g, 15 mmol). The flask was heated to 100 °C for 1.5 h. The solution was then allowed to cool to room temperature and quenched by adding a solution of sodium acetate (8.3 g) in water (15 mL). The reaction mixture was then extracted with dichloromethane, washed with water and brine and subsequently dried over anhydrous sodium sulfate. The solvent was evaporated in vacuo. The product was purified by column chromatography (dichloromethane: hexane (1:4)). The product was obtained as a dark brown resin weighing 1.19 g (60%). IR*_Vmax_* (ATR): 3077, 2950 (C-H), 1622 (C=C), 1520, 1445 (C=C) cm^−1^. ^1^H NMR (400 MHz, CDCl_3_) δ 1.77 (d, *J =* 7.32 Hz, 6 H, 2 × CH_3_), 4.52–4.62 (m, 1 H, CH), 7.50–7.55 (m, 2 H, 2 × ArH), 7.58–7.65 (m, 2 H, 2 × ArH), 8.53 (d, *J =* 9.16 Hz, 2 H, 2 × ArH), 8.93 (d, *J =* 9.16 Hz, 2 H, 2 × ArH), 11.46 (s, 1 H, CHO). ^13^C NMR (101 MHz, CDCl_3_) ppm 21.3 (CH_3_), 22.1 (CH), 121.6 (CH), 123.5 (CH), 124.2 (CH), 125.0 (CH), 125.6 (CH), 125.7 (CH), 127.3 (CH), 128.3 (CH), 128.7, 129.1 (CH), 129.2 (CH), 131.8, 135.2 (CH), 146.1, 195.4 (CHO). HRMS (APCI) calculated for C_18_H_17_O [M^+^ + H] 249.1279: found 249.1276.

### 3.2. 9-(2-Methoxyvinyl)anthracene (***17***)

Potassium *tert*-butoxide (1 M solution in THF, 13 mL) was added dropwise to a suspension of (methoxymethyl)triphenylphosphonium chloride (4.46 g, 13 mmol) in THF (25 mL) under nitrogen at 0 °C. The resultant mixture was stirred for 1 h. After 1 h, a solution of anthraldehyde (10 mmol) in THF (20 mL) was added. The reaction was stirred at 0 °C for 5 min and then allowed to heat to room temperature, after which the reaction was allowed to stir for 2 h. The reaction was then filtered through a short column of silica and eluted with diethyl ether. Solvent was removed in vacuo. The product was purified by column chromatography (hexane: dichloromethane (3:2)) to afford a mixture of *E/Z* methylenol ethers. The product was obtained as an orange resin weighing 1.83 g (60%). IR*_Vmax_* (ATR): 2981, 2924 (C-H), 1622 (C=C), 1524, 1445 (C=C) cm^−1^. ^1^H NMR (400 MHz, CDCl_3_) δ 3.66 (s, 3 H, *Z* CH_3_), 3.96 (s, 3 H, *E* CH_3_), 6.04 (d, *J =* 6.71 Hz, 1 H, *Z* =CH), 6.46 (d, *J =* 13.43 Hz, 1 H, *E* =CH), 6.61 (d, *J =* 7.32 Hz, 1 H, *Z* =CH), 6.81 (d, *J =* 12.82 Hz, 1 H, E =CH), 7.46–7.58 (m, 6 H), 7.99–8.09 (m, 3 H), 8.36–8.49 (m, 3 H). ^13^C NMR (101 MHz, CDCl_3_) ppm 56.5, 59, 98.9, 101.3, 124.9, 125.0, 125.1, 125.7, 126.1, 126.2, 126.7, 128.4, 128.5, 128.6, 128.7, 129.4, 129.6, 130.1, 130.5, 131.4, 131.5, 133.6, 133.8, 148.5, 152.8. HRMS (APCI) calculated for C_17_H_15_O [M^+^ + H] 235.1123: found 235.1115.

### 3.3. General Procedure for the Preparation of Nitrovinyl Anthracene Derivatives (***19h-m***, ***20a***, ***20b***)

Piperidinium acetate (1.5 g, 10.3 mmol) was added to a solution of 9-anthraldehyde (2 g, 9.7 mmol) in the appropriate nitroalkane (15 mL). Piperidinium acetate was prepared from piperidine (6.6 mL) and acetic acid (3 mL). The solution was heated at 90 °C for 1.5 h under nitrogen gas. After one hour, the reaction was cooled to room temperature and poured onto ice-cold H_2_O (100 mL). The resultant mixture was extracted into DCM and washed with brine, and the organic layers were combined, dried over Na_2_SO_4_ and the solvent removed in vacuo. The product was recrystallised from an appropriate solvent.

#### 3.3.1. (*E*)-9-Bromo-10-(2-Nitrovinyl)Anthracene (**19h**)

(*E*)-9-Bromo-10-(2-nitrovinyl) anthracene was prepared from 10-bromoanthracene-9-carbaldehyde (1.7 mmol, 0.5 g) and nitromethane (5.1 mL) according to the general procedure above. The product was purified by column chromatography using a mobile phase of dichloromethane: hexane, 1:2, and obtained as an orange solid weighing 311 mg (56%), Mp. 227–230 °C. IR*_Vmax_* (ATR): 3101, 3048 (C-H), 1626, 1508, (C=C), 1543, 1347 (NO_2_), 1251 (C-N) cm^−1^. ^1^H NMR (400 MHz, DMSO-*d*_6_) δ 7.71 (br. s., 2 H, 2 × ArH), 7.80 (br. s., 2 H, 2 × ArH), 7.92 (d, *J =* 13.27 Hz, 1 H, H1′), 8.31 (d, *J =* 7.46 Hz, 2 H, 2 × ArH), 8.54 (d, *J =* 7.46 Hz, 2 H, 2 × ArH), 8.94 (d, *J =* 13.68 Hz, 1 H, H2′). ^13^C NMR (101 MHz, DMSO-*d*_6_) ppm 109.5, 126.0 (CH), 127.4 (CH), 127.5 (CH), 128.2 (CH), 129.4 (CH), 129.6 (CH), 135.1, 137.6, 144.5, 148.3 (C2′). HRMS (APCI) calculated for C_16_H_11_Br^79^NO_2_ [M^+^ + H] 327.9973: found 327.9966.

#### 3.3.2. (*E*)-9-Methyl-10-(2-Nitrovinyl)Anthracene (**19i**)

(*E*)-9-Methyl-10-(2-nitrovinyl) anthracene was prepared from 10-methylanthracene-9-carbaldehyde (2.2 mmol, 0.5 g) and nitromethane (6.6 mL) according to the general procedure above. The product was purified by column chromatography (dichloromethane: hexane, 1:2). The product was recrystallised from methanol and diethyl ether as a bright orange solid weighing 175 mg (30%), Mp. 140–142 °C. IR*_Vmax_* (ATR): 3100, 2981 (C-H), 1624 (C=C), 1549, 1332 (NO_2_), 1504, 1461 (C=C) cm^−1^. ^1^H NMR (400 MHz, CDCl_3_) δ 3.08 (s, 3 H, CH_3_), 7.45 (d, *J =* 13.43 Hz, 1 H, H1′), 7.51–7.61 (m, 4 H, 4 × ArH), 8.08–8.17 (m, 2 H, 2 × ArH), 8.23–8.34 (m, 2 H, 2 × ArH), 8.88 (d, *J =* 14.04 Hz, 1 H, H2′). ^13^C NMR (101 MHz, CDCl_3_) ppm 14.6 (CH_3_), 121.7, 124.9 (CH), 125.4 (CH), 125.5 (CH), 126.8 (CH), 129.4, 129.6, 134.6, 136.2 (CH), 142.5 (C2′). HRMS (APCI) calculated for C_17_H_14_NO_2_ [M^+^ + H] 264.1019: found 264.1006.

#### 3.3.3. (*E*)-9-(2-Nitrovinyl)-10-Phenylanthracene (**19j**)

(*E*)-9-(2-Nitrovinyl)-10-phenyl anthracene was prepared from 10-phenylanthracene-9-carbaldehyde (1.7 mmol, 0.5 g) and nitromethane (5.1 mL) according to the general procedure above. The product was purified by column chromatography (dichloromethane: hexane, 1:1). The product was recrystallised from methanol and diethyl ether as an orange solid weighing 332 mg (60%), Mp. 119–121 °C. IR _Vmax_ (ATR): 3094, 2981 (C-H), 1630 (C=C), 1510, 1437 (C=C), 1542, 1331 (NO_2_) cm^−1^. ^1^H NMR (400 MHz, CDCl_3_) δ 7.43 (t, *J =* 7.02 Hz, 4 H, 4 × ArH), 7.55–7.66 (m, 6 H, 5 × ArH and H1′), 7.73 (d, *J =* 8.55 Hz, 2 H, 2 × ArH), 8.25 (d, *J =* 9.16 Hz, 2 H, 2 × ArH), 9.08 (d, *J =* 14.04 Hz, 1 H, H2′). ^13^C NMR (101 MHz, CDCl_3_) ppm 123.3, 124.4 (CH), 125.5 (CH), 127.1 (CH), 127.9 (CH), 128.5 (CH), 129.6, 129.9, 130.9 (CH), 136.1 (CH), 138.1, 140.9, 142.9 (C2′). HRMS (APCI) calculated for C_22_H_16_NO_2_ [M^+^ + H] 326.1181: found 326.1177.

#### 3.3.4. (*E*)-1,8-Dichloro-10-(2-Nitrovinyl)Anthracene (**19k**)

(*E*)-1,8-Dichloro-10-(2-nitrovinyl) anthracene was prepared from 4,5-dichloroanthracene-9-carbaldehyde (1.8 mmol, 0.5 g) and nitromethane (5.1 mL) according to the general procedure above. The product was purified by column chromatography (dichloromethane: hexane, 1:1) and obtained as a yellow solid weighing 58 mg (10%), Mp. 206–209 °C. IR*_Vmax_* (ATR): 2981, 2889 (C-H), 1631 (C=C), 1510, 1437 (C=C), 1543, 1361 (NO_2_) cm^−1^. ^1^H NMR (400 MHz, CDCl_3_) δ 7.40–7.63 (m, 3 H, 2 × ArH and H1′), 7.71 (d, *J =* 7.32 Hz, 2 H, 2 × ArH), 8.08 (d, *J =* 8.55 Hz, 2 H, 2 × ArH), 8.91 (d, *J =* 14.04 Hz, 1 H, H2′), 9.47 (s, 1 H, H9). ^13^C NMR (101 MHz, CDCl_3_) ppm 123.6 (CH), 124.3 (CH), 126.3, 126.5, 127.5 (CH), 129.0, 130.7, 133.5, 135.3 (CH), 143.6 (C2′). HRMS (APCI) calculated for C_16_H_9_^35^Cl_2_NO_2_ [M^+^] 317.0010: found 317.0009.

#### 3.3.5. (*E*)-9-Ethyl-10-(2-Nitrovinyl)Anthracene (**19l**)

(*E*)-9-Ethyl-10-(2-nitrovinyl)anthracene was prepared from 10-ethylanthracene-9-carbaldehyde (4.4 mmol, 1.1 g) and nitromethane (12 mL) according to the general procedure above. The product was purified by column chromatography dichloromethane: hexane (1:1), and recrystallised from methanol and diethyl ether as an orange solid weighing 733 mg (60%), Mp. 130–134 °C. IR*_Vmax_* (ATR): 3087, 2976 (C-H), 1629 (C=C), 1554, 1338 (NO_2_), 1539, 1442 (C=C) cm^−1^. ^1^H NMR (400 MHz, CDCl_3_) δ 1.49 (t, *J =* 7.63 Hz, 3 H, CH_3_), 3.68 (q, *J =* 7.32 Hz, 2 H, CH_2_), 7.46–7.66 (m, 5 H, 4 × ArH, H1′), 8.14–8.25 (m, 2 H, 2 × ArH), 8.30–8.42 (m, 2 H, 2 × ArH), 8.98 (d, *J =* 13.43 Hz, 1 H, H2′). ^13^C NMR (101 MHz, CDCl_3_) ppm 15.5 (CH_3_), 21.6 (CH_2_), 122.0, 124.3 (CH), 125.1 (CH), 125.2 (CH), 125.7 (CH), 125.7 (CH), 126.8 (CH), 127.5 (CH), 128.8, 129.2 (CH), 129.8, 136.4 (CH), 140.9, 142.6 (C2′). HRMS (APCI) calculated for C_18_H_16_NO_2_ [M^+^ + H] 278.1181: found 278.1178.

#### 3.3.6. (*E*)-9-Isopropyl-10-(2-Nitrovinyl)Anthracene (**19m**)

(*E*)-9-Isopropyl-10-(2-nitrovinyl)anthracene was prepared from 10-isopropylanthracene-9-carbaldehyde (4.4 mmol, 1.1 g) and nitromethane (12 mL) according to the general procedure above. The product was purified by column chromatography dichloromethane: hexane (1:1) and isolated from methanol and diethyl ether as a dark red resin weighing 873 mg (68%). IR*_Vmax_* (ATR): 3087, 2962 (C-H), 1628 (C=C), 1552, 1333 (NO_2_), 1514, 1443 (C=C) cm^−1^. ^1^H NMR (400 MHz, CDCl_3_) δ 1.80 (m, 6 H, 2 × CH_3_), 4.52–4.71 (m, 1 H, CH), 7.43–7.62 (m, 5 H, 4 × ArH, H1′), 8.12–8.24 (m, 2 H, 2 × ArH), 8.47–8.63 (m, 2 H, 2 × ArH), 8.99 (dd, *J =* 13.43, 1 H, H2′). ^13^C NMR (101 MHz, CDCl_3_) ppm 22.9 (CH_3_), 28.8 (CH), 122.5, 124.3 (CH), 125.1, 125.4 (CH), 125.6 (CH), 125.8 (CH), 126.4 (CH), 127.5 (CH), 129.0, 129.0, 129.2 (CH), 129.9, 130.0, 136.5 (CH), 142.8 (C2′), 144.7. HRMS (APCI) calculated for C_19_H_18_NO_2_ [M^+^ + H] 292.1338: found 292.1334.

#### 3.3.7. 9-(3-Nitroallyl)Anthracene (**20a**)

9-(3-Nitroallyl)anthracene was prepared from 2-(anthracen-9-yl)acetaldehyde (6.7 mmol, 1.5 g) and nitroethane (15 mL) according to the general procedure above. The product was purified by column chromatography (eluent 3:1 dichloromethane: hexane) and obtained as a dark orange oil weighing 566 mg (32%). IR*_Vmax_* (ATR): 3089, 2928 (C-H), 1667 (C=C), 1551, 1356 (NO_2_), 1514, 1444 (C=C) cm^−1^. ^1^H NMR (400 MHz, CDCl_3_) δ 4.58 (d, *J =* 6.10 Hz, 2 H, CH_2_), 6.62 (dd, *J =* 13.43, 1.83 Hz, 1 H, H3′), 7.49–7.61 (m, 4 H, 4 × ArH), 7.66–7.77 (m, 1 H, H2′), 8.07 (d, *J =* 8.55 Hz, 4 H, 4 × ArH), 8.49 (s, 1 H, H10). ^13^C NMR (101 MHz, CDCl_3_) ppm 27.0 (CH_2_), 123.3 (CH), 125.2 (CH), 126.8 (CH), 127.8 (CH), 129.5 (CH), 129.8, 131.5, 140.7 (C2′), 141.1. HRMS (APCI) calculated for C_17_H_14_NO_2_ [M^+^ + H] 264.1025: found 264.1023.

#### 3.3.8. 9-(3-Nitrobut-2-en-1-yl)Anthracene (**20b**)

9-(3-Nitrobut-2-en-1-yl)anthracene was prepared from 2-(anthracen-9-yl)acetaldehyde (6.7 mmol, 1.5 g) and nitroethane (15 mL) according to the general procedure above. The product was purified by column chromatography (eluent 3:1 dichloromethane: hexane) and obtained as a brown oil weighing 372 mg (20%). IR*_Vmax_* (ATR): 3088, 2951 (C-H), 1667 (C=C), 1545, 1360 (NO_2_), 1513, 1473 (C=C) cm^−1^. ^1^H NMR (400 MHz, CDCl_3_) δ 2.54 (s, 3 H, CH_3_), 4.53 (d, *J =* 6.71 Hz, 2 H, CH_2_), 7.49–7.60 (m, 5 H, 4 × ArH and H2′), 8.05–8.13 (m, 4 H, 4 × ArH), 8.47 (s, 1 H, H10). ^13^C NMR (101 MHz, CDCl_3_) ppm 13.1 (CH_3_), 26.7 (CH_2_), 123.4 (CH), 125.1 (CH), 126.5 (CH), 127.5 (CH), 128.5, 129.6 (CH), 129.8, 131.6, 134.9 (C2′). HRMS (APCI) calculated for C_18_H_16_NO_2_ [M^+^ + H] 278.1181: found 278.1177.

### 3.4. 9-(2-Nitroethyl)Anthracene (***22***)

Sodium borohydride (60 mg, 1.6 mmol) was added to a solution of (*E*)-9-(2-nitrovinyl)anthracene (100 mg, 0.4 mmol) in dichloromethane (10 mL) and isopropanol (2 mL). The reaction mixture was stirred at room temperature for 24 h and neutralised using 1 M HCl. The solution was extracted with dichloromethane, dried with sodium sulfate and the solvent removed in vacuo. The product was recrystallised from methanol and diethyl ether as orange crystals weighing 85 mg (85%), Mp. 147–149 °C. IR*_Vmax_* (ATR): 3053, 2974 (C-H), 1622, 1493 (C=C), 1546, 1377 (NO_2_), 1137 (C-N) cm^−1^. ^1^H NMR (400 MHz, CDCl_3_) δ 4.35–4.44 (m, 2 H, CH_2_), 4.70–4.78 (m, 2 H, CH_2_), 7.48–7.56 (m, 2 H, 2 × ArH), 7.57–7.65 (m, 2 H, 2 × ArH), 8.06 (d, *J =* 8.55 Hz, 2 H, 2 × ArH), 8.25 (d, *J =* 9.16 Hz, 2 H, 2 × ArH), 8.46 (s, 1 H, C10). ^13^C NMR (101 MHz, CDCl_3_) ppm 26.1 (C2′), 74.7 (C1′), 122.9 (CH), 125.2 (CH), 126.2, 126.9 (CH), 127.8, 129.6 (CH), 129.9, 131.5. HRMS (APCI) calculated for C_16_H_12_NO_2_ [M^+^ − H] 250.0874: found 250.0874.

### 3.5. 2-((10-Chloroanthracen-9-yl)Methylene)Malononitrile (***23b***)

One drop of piperidine was added to a solution of 10-chloro-9-anthraldehyde (0.458 g, 1.91 mmol) and malonitrile (0.372 g. 6.11 mmol) in ACN (50 mL). The solution was heated at 90 °C for 0.5 h. The solution was then concentrated in vacuo, and the residual solid was dissolved in DCM and washed with HCl (10%), water and brine. The organic phases were combined and dried over sodium sulfate and the solvent removed in vacuo. The product was obtained as orange crystals weighing 495 mg (90%), Mp. 196–198 °C. IR*_Vmax_* (ATR): 3088, 2965 (C-H), 2228 (CN), 1621 (C=C), 1574, 1445 (C=C) cm^−1^. ^1^H NMR (400 MHz, CDCl_3_) δ 7.68–7.76 (m, 4 H, 4 × ArH), 7.91–7.97 (m, 2 H, 2 × ArH), 8.61–8.71 (m, 2 H, 2 × ArH), 8.91 (s, 1 H, C1′). ^13^C NMR (101 MHz, CDCl_3_) ppm 93.4 (C2′), 111.1 (CN), 112.7 (CN), 122.9, 124.2 (CH), 126.2 (CH), 127.2, 127.4 (CH), 128.4, 128.5 (CH), 129.3, 160.2 (C1′). HRMS (APCI) calculated for C_18_H_9_N_2_Cl^35^ [M^+^] 288.0454: found 288.0466.

### 3.6. 4-(Anthracen-9-yl)-5-Methyl-1H-1,2,3-Triazole (***30b***)

Sodium azide (2 mmol) was added to a stirred solution of (*E*)-9-(2-nitroprop-1-en-1-yl)anthracene (1 mmol, 0.25 g) in DMSO (2 mL). The reaction was stirred at 90 °C for 90 min. After allowing the reaction to cool to room temperature, deionised water (10 mL) was added, and the aqueous mix was extracted into diethyl ether. The organic layer was washed with water and brine and dried over sodium sulfate. The solvent was removed in vacuo to afford a crude product. The product was purified by column chromatography (gradient dichloromethane–methanol) and recrystallised from ethanol as a yellow solid weighing 117 mg (45%), Mp. 231–232 °C. IR*_Vmax_* (ATR): 3082, 3026 (C-H), 1623, 1587 (C=C), 1249 (C-N) cm^−1^. ^1^H NMR (400 MHz, DMSO-*d*_6_) d 2.01 (s, 3 H, CH_3_), 7.40–7.63 (m, 6 H, 6 × ArH), 8.17 (d, *J =* 8.29 Hz, 2 H, 2 × ArH), 8.76 (s, 1 H, H10). ^13^C NMR (101 MHz, DMSO-*d*_6_) ppm 9.3 (CH_3_), 125.5 (CH), 126.4 (CH), 128.6 (CH), 130.5, 130.8. HRMS (APCI) calculated for C_17_H_14_N_3_ [M^+^ + H] 260.1188: found 260.1181.

### 3.7. 4-(Anthracen-9-yl)-4,5-Dihydro-1H-1,2,3-Triazole (***31***)

Sodium azide (2 mmol) was added to a stirred solution of (*E*)-9-(2-nitrovinyl)anthracene (1 mmol, 0.25 g) in DMSO (2 mL). The reaction was stirred at 90 °C for 12 h. After allowing the reaction to cool to room temperature, deionised water (10 mL) was added, and the aqueous mix was extracted into diethyl ether. The organic layer was washed with water and brine and dried over sodium sulfate. The solvent was removed in vacuo to afford a crude product. The product was purified by column chromatography (gradient dichloromethane–methanol) and recrystallised from ethanol as yellow crystals weighing 222 mg (90%), Mp. 266–268 °C. IR*_Vmax_* (ATR): 3054, 2920 (C-H), 1614, 1488 (C=C), 1220 (C-N) cm^−1^. ^1^H NMR (400 MHz, DMSO*-d_6_*) δ 4.64 (d, *J =* 4.88 Hz, 2 H, CH_2_), 6.68 (t, *J =* 5.19 Hz, 1 H, CH), 7.45–7.70 (m, 4 H, 4 × ArH), 8.11 (d, *J =* 8.55 Hz, 2 H, 2 × ArH), 8.31 (d, *J =* 9.16 Hz, 2 H, 2 × ArH), 8.56 (s, 1 H, H10), 11.40 (s, 1 H, NH). ^13^C NMR (101 MHz, DMSO*-d_6_*) ppm 24.0 (CH_2_), 124.3 (CH), 125.2 (CH), 126.2 (CH), 126.3 (CH), 129.0 (CH), 129.3, 129.7, 131.1, 147.9 (CH). HRMS (APCI) calculated for C_16_H_13_N_3_ [M^+^] 247.1109: found 247.1118.

### 3.8. Anthracen-9-yl(Azepan-1-yl)Methanone (***33c***)

Chloro-1-methylpyridinium iodide (0.76 g, 3 mmol) was added to a suspension of anthracene-9-carboxylic acid (0.22 g, 1 mmol) in anhydrous dichloromethane (10 mL). The solution turned yellow and was stirred for a further 5 min. Homopiperidine (0.113 mL, 1 mmol) was added to this followed by the addition of trimethylamine (0.7 mL, 5 mmol). The reaction mixture was stirred at room temperature for 1 h and was then diluted with 10% hydrochloric acid (10 mL) followed by extraction with dichloromethane (10 mL). The organic layer was washed with 10% sodium hydroxide solution, water and brine. The solvent was removed in vacuo, and the residue was purified by column chromatography (eluent: dichloromethane) and recrystallised from ethanol as a brown solid weighing 261 mg (86%), Mp. 134–137 °C. IR*_Vmax_* (ATR): 3048, 2929 (C-H), 1615, 1486.60 (C=C), 1263 (C-N) cm^−1^. ^1^H NMR (400 MHz, CDCl_3_) δ 1.31–1.43 (m, 2 H, CH_2_), 1.47–1.59 (m, 2 H, CH_2_), 1.75 (dt, *J =* 11.75, 6.03 Hz, 2 H, CH_2_), 1.94–2.12 (m, 2 H, CH_2_), 3.09 (t, *J =* 6.10 Hz, 2 H, CH_2_), 3.95–4.06 (m, 2 H, CH_2_), 7.42–7.57 (m, 4 H, 4 × ArH), 7.90–8.06 (m, 4 H, 4 × ArH), 8.45 (s, 1 H, H10). ^13^C NMR (101 MHz, CDCl_3_) ppm 26.5 (C5″/C4″), 28.1 (C5″/C4″), 28.3 (C3″/C6″), 28.8 (C3″/C6″), 45.1 (C7″/C2″), 49.0 (C7″/C2″), 125.0 (CH), 125.4 (CH), 126.5 (CH), 127.3 (CH), 127.5, 128.6 (CH), 131.2, 131.7, 170.1 (C1′). HRMS (APCI) calculated for C_21_H_22_NO [M^+^ + H] 304.1701: found 304.1688.

## 4. Biochemistry

### 4.1. Materials

The MUTU-I (c179), DG-75, BJAB and Ramos BL cell lines were gifts from Dr. Dermot Walls (School of Biotechnology, Dublin City University, Dublin, Ireland). The EBV-transformed CLL PGA-1 (M-IGVH, good prognosis) and HG-3 (UM-IGVH, poor prognosis) cell lines were provided by Professor Anders Rosén (Linköping University, Linköping, Sweden) [116]. The MCF-7 human breast carcinoma cells were purchased from the European Collection of Animal Cell Cultures (ECACC). The MDA-MB-231 human breast carcinoma cell line was a gift from Dr. Susan McDonnell, School of Chemical and Bioprocess Engineering, University College Dublin. Other cell lines were sourced as follows: HL-60 promyelocytic leukaemia cells (cat. no. 98070106; ECACC, Salisbury, UK) and HeLa cervical cancer cells (cat. no. 93021013; ECACC). alamarBlue was obtained from BioSource, Nivelles, Belgium, and foetal bovine serum (FBS) was sourced from Invitrogen, U.K. RPMI 1640 medium, DMEM, HEPES, sodium pyruvate, Gentamycin (G418) and glutamine were sourced from Thermo Fisher Scientific, Dun Laoghaire, Ireland. Cell culture consumables were obtained from Greiner Bio-One Ltd., Stonehouse, U.K., while all other reagents used were obtained from Sigma-Aldrich (now Merck), Arklow, Ireland.

### 4.2. Cell Culture

The MUTU-I and chemoresistant DG-75 BL cell lines were grown in RPMI 1640 (Glutamax) medium containing phenol red and supplemented with 10% (*v*/*v*) FBS, L-glutamine (2 mM), 50 U/mL penicillin and 50 μg/mL streptomycin. The MUTU-I cell line also required supplements of sodium pyruvate (100 mM), alpha-thioglycerol (5 mM in phosphate buffered saline (PBS) with 20 μM bathocuprione disulfonic acid) and HEPES (1 mM). The CLL PGA-1 and HG-3 and HL-60 leukaemia cell lines were grown in RPMI-1640 (Glutamax; Thermo Fisher Scientific, Inc., Dun Laoghaire, Dublin, Ireland) medium supplemented with 10% (*v*/*v*) FBS, 50 μg/mL streptomycin and 50 U/mL penicillin and seeded at a density of 2 × 10^5^ cells/mL. MCF-7 cells were maintained in Eagle’s minimum essential medium (MEM) supplemented with 10% (*v*/*v*) foetal bovine serum (FBS), L-glutamine (2 mM), penicillin/streptomycin (100 μg/mL) and non-essential amino acids (1% (*v*/*v*). The MDA-MB-231 cells were maintained in Dulbecco’s modified Eagle’s medium (D-MEM) supplemented with FBS, (10% (*v*/*v*)), l-glutamine (2 mM) and penicillin/streptomycin (100 μg/mL). HeLa cervical cancer cells were grown in DMEM supplemented with 10% (*v*/*v*) FBS and 50 μg/mL penicillin/streptomycin and seeded at a density of 5 × 10^4^ cells/mL. All cell cultures were maintained at 37 °C under a humidified atmosphere of 5% CO_2_/95% O_2_ and were passaged at least twice weekly depending on their levels of confluency.

### 4.3. AlamarBlue Cell Viability Assay

Cells were seeded at a density of 2.5 × 10^4^ cells/well (MCF-7, MDA-MB-231 cells), 1 × 10^4^ cells/well (HL-60 cells), 1–5 × 10^4^ cells per well (BL MUTU-I, DG-75 cells) and 2 × 10^5^ cells/well [HG-3, PGA-1 (CLL cells)], Ramos, BJAB (BL cells) (200 μL per well), in 96-well plates. Cells were treated with the desired drug concentration for the appropriate time for each cell type, and the samples were incubated as required. alamarBlue (20 μL) was then added to each well and the samples further incubated in the dark at 37 °C for 4 h. Wells containing only reagent and cell culture medium in the absence of cells were used as blank controls. Ethanol was used as a vehicle, and cells were treated with 1% ethanol (*v*/*v*) in all experiments. The 96-well plates were analysed on a fluorescence plate reader (SpectraMax Gemini, Molecular Devices) (emission and excitement wavelengths of 590 nm and 544 nm, respectively), and the fluorescence was recorded. The decrease in cell viability was then calculated with reference to the vehicle samples (100% viability). EMEM medium with the addition of alamarBlue was used as a blank. Vehicle-treated cells were considered to be 100% viable, and the viabilities of each compound was calculated accordingly. The transformed data (Final Concentration = Log (Final Concentration)) was used to plot a non-linear, sigmoidal dose–response curve, and the concentration of drug resulting in a 50% reduction in cell survival (IC_50_ values) was obtained using the software package Prism (GraphPad Software, Inc., La Jolla, CA, USA). Taxol (10 μM) was used as an internal standard and resulted in a 90% cytotoxicity in each of the cell lines. All biochemical assays were performed in triplicate on at least three independent occasions, and the mean values were determined.

### 4.4. Generation of Human Peripheral Blood Mononuclear Cells (PBMCs)

Peripheral blood was obtained from healthy donors (n = 2) after informed consent was received. The blood was then placed into a 50 mL falcon tube and diluted with an equal volume of phosphate-buffered saline (PBS). PBMCs were isolated using density gradient centrifugation using LymphoPrep as described previously [117] Approval for this study was obtained from the School of Pharmacy and Pharmaceutical Sciences Trinity College Dublin Research Ethics Committee (2020-06-01-MS).

### 4.5. Annexin V/PI Apoptotic Assay

The BL cells DG-75 and MUTU-I (750,000) were treated at 37 °C with either vehicle (0.1% (*v*/*v*) EtOH), the nitrostyrene compounds **11i, 11h, 11g, 11l, 11j** and **11k** at 10 μM concentration or the positive control drug taxol (10 μM) and incubated for 24 h (MUTU-1) or 48 h (DG-75). The CLL cells PGA1 and HG3 (1 × 10^6^ cells/mL) were treated at 37 °C with either vehicle (1% (*v*/*v*) DMSO) or the nitrostyrene compounds **19a** (10 μM, 5 μM and 1 μM) and **19g, 19i**, **19l** and **19m** (10 μM and 1 μM) for 48 h. The cells were then harvested by centrifugation at 400× *g* in a temperature-controlled Sorvall centrifuge and rinsed with 0.5 mL of Ca^2+^ Annexin-V-binding buffer (0.1 M HEPES, pH 7.4; 0.14 M NaCl; 25 mM CaCl_2_). The samples were resuspended in 50 µL FITC Annexin V (diluted 1:33 in Ca^2+^ Annexin V-binding buffer), incubated on ice for 10 min (protected from light), washed with Annexin-V-binding buffer and re-suspended in 500 µL of propidium iodide (PI) solution (0.5 µg/mL). The samples were analysed within 1 h using a BD Accuri C6 flow cytometer counting 10,000 cells and analysed using the FlowJo software (FlowJo LLC, Ashland, OR, USA) package.

### 4.6. Inhibitor Studies

The HG-3 and PGA-1 CLL cells (5 × 10^4^ cells/mL) were pre-treated at 37 °C with either 5 mM N-acetylcysteine (NAC) for 1 h or 40 μM caspase inhibitor (Z-VAD-FMK; MBL International, Co., Woburn, MA, USA) for 4 h prior to treatment with (11 h at 5 μM) for 48 h. Annexin V/PI FACS analysis was then carried out as described above in Section 4.5.

### 4.7. X-ray Experimental Procedure

Data for **19f** were measured on a Bruker D8 Quest ECO device using Mo Ka radiation (λ = 0.71073 Å) with an Oxford Cryostream low-temperature device, and data for **30a** were collected on a Bruker APEX DUO device using Cu Kα radiation (λ = 1.54178 Å). Each sample was mounted on a MiTeGen cryoloop, and data were collected at 100(2) K. The Bruker APEX [118] software was used to collect and reduce data. Absorption corrections were applied using SADABS [119]. Structures were solved with the SHELXT structure solution program [120] using Intrinsic Phasing. All were refined using the least squares method on F^2^ with SHELXL [121]. All non-hydrogen atoms were refined anisotropically. Hydrogen atoms were assigned to calculated positions using a riding model with appropriately fixed isotropic thermal parameters. Molecular graphics were generated using OLEX2 [122]. Crystal data, details of data collection and refinement are given in Appendix A.

Compound **30a** was a weakly diffracting sample, especially at high angle, and it had two independent molecules in the asymmetric unit. One complete anthracene-triazole molecule was disordered over two locations (50% occupancy) and modelled with displacement restraints (SIMU).

Crystallographic data for the structures in this paper have been deposited with the Cambridge Crystallographic Data Centre as supplementary publication nos. 2171050-2171051. Copies of the data can be obtained, free of charge, on application to CCDC, 12 Union Road, Cambridge CB2 1EZ, UK, (fax: +44-(0)1223-336033 or e-mail: deposit@ccdc.cam.ac.uk).

### 4.8. Computational Overlay Study

For the MOE 2022.02 work, all the compounds were opened in a database viewer. The compounds were washed with default values and explicit hydrogens were added. For each compound, MMFF94x partial charges were calculated, and each was minimised to a gradient of 0.001 kcal/mol/Å. The compounds were then overlaid individually on 3D structure of maprotiline using flexible alignment on MOE with default values. In the fastROCS [86] 3D similarity study, 200 conformations were generated for each compound using OpenEye Omega [84,123]. This conformation file was then used to search for compound conformations that map to a low-energy conformation of maprotiline. The output hit list was sorted according to the combined Tanimoto scores.

## 5. Conclusions

The treatment of hematological malignancies such as CLL is evolving as the underlying mechanisms of disease are understood and new immune pathways are discovered. Patients with CLL do not need treatment with chemotherapy until they become symptomatic or display evidence of rapid progression of the disease. The drug treatments available for patients with CLL has improved considerably and now include very effective oral targeted therapies (such as ibrutinib **2**, idelalisib **3** and venetoclax **4**). While immunotherapies such as obinutuzumab have proven successful in treating CLL, more research is required to optimise the current chemotherapies and immunotherapies for blood cancers, the combination and sequencing of treatments and the development of personalised and targeted agents. Many recent developments have been reported in the targeted clinical treatment of CLL; however, the discovery of novel therapeutic agents that are designed to be effective in acquired disease resistance and provide a curative treatment rather than maintenance alone is required.

We investigated the synthesis, structure−activity relationship (SAR) studies and biological activity of a series of nitrostyrenes and nitrovinylanthracenes in a phenotypic drug discovery approach. Potential preclincial applications of the novel anthracene nitrostyrene compounds in BL and CLL (a more translational but related B-cell malignancy) were identified. A library of 58 anthracene-based compounds structurally related to maprotiline were initially evaluated for antiproliferative activity in the BL EBV^−^MUTU-I (chemosensitive) and EBV^+^DG-75 (chemoresistant) cell lines. The selected (*E*)-9-(2-nitrovinyl)anthracenes demonstrated potent antiproliferative activity in the CLL cell lines, with IC_50_ values of 0.17 μM (HG-3) and 1.3 μM (PGA-1) for compound **19g**, superior to the chemotherapeutic drug fludarabine with IC_50_ values of 28.1 μM (HG-3) and 32.0 μM (PGA-1). The pro-apoptotic effects of the most potent compounds **19a, 19g, 19i, 19l** and **19m** were demonstrated in both the CLL cell lines. The mechanism of cell death in these cell lines was identified as apoptotic, and the lead compounds elicited significant apoptotic effects which were comparable to taxol in the BL cell lines MUTU-I and DG-75 and fludarabine in the CLL cell lines HG-3 and PGA-1. The nitroalkenes reported in this study are attractive substrates for Michael addition reactions with nucleophiles present in biological systems. A molecular modelling study demonstrated close correspondence between overlays of these compounds with maprotiline with shared molecular features; however, correlation with cellular activity was inconclusive. The (*E*) nitrostyrene and (*E*)-9-(2-nitrovinyl)anthracene series of compounds offer potential for further development as novel chemotherapeutics for chronic lymphocytic leukaemia (CLL) and suggest the suitability of this group of selected (*E*)-9-(2-nitrovinyl)anthracenes compounds for further preclinical development.

## Data Availability

Data are contained within the article and Appendix A.

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
