# Peer review of "Synthesis and Pro-Apoptotic Effects of Nitrovinylanthracenes and Related Compounds in Chronic Lymphocytic Leukaemia (CLL) and Burkitt’s Lymphoma (BL)"

_molecules, 2023, doi:10.3390/molecules28248095_

Round 1

Reviewer 1 Report

Comments and Suggestions for Authors

The work describes the synthesis of 58 compounds, theoretical analysis of their properties, and preliminary biological research. In my opinion, the work cannot be accepted in this form. Below are my comments:

1- the work is too long, it describes the synthesis of various groups of compounds (styrene derivatives and anthracene derivatives), the Authors should divide it into at least two independent works,

2- in the case of inactive compounds, the Authors should consider describing their synthesis and characterization in a chemical journal

3 -only a few active derivatives were identified in the work and for these compounds, the Authors should deepen biological research and add e.g.: caspases activity, analysis of cell cycle, ROS production, toxicity towards normal cells....

Author Response

Reviewer comments: The work describes the synthesis of 58 compounds, theoretical analysis of their properties, and preliminary biological research. In my opinion, the work cannot be accepted in this form. Below are my comments:

Author response:      We thank the reviewer for their feedback. We have endeavoured to revise and modify the manuscript which we hope will be acceptable. Our responses are provided in detail below:

Reviewer comments 1:- the work is too long, it describes the synthesis of various groups of compounds (styrene derivatives and anthracene derivatives), the Authors should divide it into at least two independent works,

Author response: We thank the reviewer for their suggestion to divide the manuscript. This work describes the synthesis, characterisation and evaluation of a series of nitrostyrene, related nitrovinylanthracenes and structural modification of these anthracenes in Burkitt’s lymphoma (BL) and chronic lymphocytic leukaemia (CLL). We initially explored the structural requirements for antiproliferative activity of a panel of 58 compounds in BL cell lines, a rare form of non-Hodgkin’s immune B-cell lymphoma. We then identified the most promising compounds for evaluation in chronic lymphocytic leukaemia (CLL) which is a more common but related B-cell malignancy. Based on our previous work in this area of design of compounds with activity against BL and CLL, our objective was to identify potential lead compounds with potent activity in the CLL cell lines, for further preclinical progression.

The work is arranged in the following sections:

  • Synthetic routes to nitrostyrenes, nitrovinylanthracenes and structural modification of these anthracenes
  • Evaluation of panel of synthesised nitrostyrenes, nitrovinylanthracenes and structurally modified anthracenes in BL cell lines; SAR study and identification of potent lead compounds for progression
  • Selection of the most potyent nitrostyrenes and nitrovinylanthracenes for evaluation in CLL cell lines; SAR study and identification of potent lead compounds for further study
  • Confirmation of proapoptotic activity of lead compounds for further preclinical evaluation
  • Identification of the nitrovinylaryl pharmacophore required for activity in CLL

We thank the reviewer for the suggestion to divide it into at least two independent works; however, with the results obtained for the panel of compounds prepared, we have presented an extensive study of the SAR and structural requirements for potent activity as antiproliferative compounds in both BL and extended to CLL. Publishing the results for BL in one work, for example, would leave sparse information for further publication of the results in CLL. The inactive compounds prepared and tested were also included in our work to present a comprehensive SAR study of this series of compounds.

Reviewer comments 2:- in the case of inactive compounds, the Authors should consider describing their synthesis and characterization in a chemical journal

Author response: We thank the reviewer for the suggestion to describe the synthesis and characterization of the inactive compounds in a chemical journal. However, as discussed above, the inactive anthracene compounds prepared and tested are included in our work as the preparation and investigation of the complete series was required to establish a comprehensive SAR analysis and to identify the lead compounds for further investigation; therefore we would like to retain data for all compounds in this manuscript, rather than remove inactive compounds to a separate chemical journal. The chemistry employed in the manuscript follows established procedures and, without biochemical evaluation, would not be suitable for a chemistry journal.

Reviewer comment 3: -only a few active derivatives were identified in the work and for these compounds, the Authors should deepen biological research and add e.g.: caspases activity, analysis of cell cycle, ROS production, toxicity towards normal cells....

Author response: We thank the reviewer for the interesting and helpful comments on the active derivatives. A comprehensive series of nitrostyrene and nitrovinylanthracene compounds were prepared  and initially evaluated in this work to complete the SAR study and to identify the most potent compounds for further biochemical study e.g. apoptosis. We did not know or anticipate in advance which compounds would possess the best activity. We wished to establish if the nitrostyrene pharmacophore was essential for antiproliferative activity of the compounds, and also if any further modifications of the aryl, anthracene, nitro or alkene sections of the molecules could be modified with the objective of enhancement of activity. From the initial screening of the 58 compounds in BL cell lines MUTU-1 and DG-75, a series of 12 nitrostyrene compounds 11a-l were identified as the most promising, with all compounds demonstrating very low cell viability in PGA-1 (0.018-7.05%) and HG-3 (0.11-11.15%) cell lines at 10 mM. Compound 11h was selected for apoptosis and ROS studies. Evaluation of the nitrovinylanthracene compounds also identified significant potency for five compounds (19a, 19g, 19i, 19l and 19m) in both BL cell lines MUTU-1 and DG-75 cells. In CLL cell lines PGA-1 and HG-3, compounds 19a, 19g, 19i, 19l and 19m were identified to be the most potent (0.5-8.3% cell viability at 10 mM in HG-3 cells and 1.21-5.1% cell viability in PGA-1 cells). These five compounds were progressed to further evaluation e.g. IC50 value determination in CLL cell lines and evaluation in human breast cancer cell lines MCF-7, MDA-MB-231 cell lines. Apoptosis investigation in CLL cell lines using Annexin /PI confirmed the proapoptotic effects of the compounds.

Additional evaluation of the effects of the compounds was performed as suggested by the reviewer. Results for reactive oxygen species (ROS) and caspase inhibition experiments are now included in this study in section 2.3 (new) and Figure 6 (new); subsequent sections are re-numbered. The biological studies will be further extended in the future but is outside the scope of this present work, which is already long.

PBMC results

The toxicity toward normal cells (PBMCs) was determined, to illustrate the low toxicity of these compounds, see text section ’ effect of nitrostyrene 11h on the viability of PBMCs on page 20 of the manuscript.

The nitrostyrene 11h was evaluated for its cytotoxic effect on healthy donor peripheral blood mononuclear cells (PBMCs) to determine the selective toxicity of compounds containing the nitrostyrene pharmacophore on malignant BL cell lines over normal blood cells. Compound 11h was evaluated at 1 μM and 10 μM concentration over a 24 h treatment time, Compound 11h demonstrated a low toxicity in PBMCs at 1 μM (74% viable cells remaining). In comparison, compound 11h induced a significant anti-proliferative effect in MUTU-I cells with 39.8% viable cells remaining at 1 μM. A similar response was observed in DG-75 cells at the higher concentration (10 μM); a potent anti-proliferative effect (0.38% viable cells remaining) was observed, in comparison to 34.1% of viable PBMCs indicating that compounds 11h is selectively toxic to these BL cell lines. This data is presented in Table S8 of the Supplementary Information.

Reviewer 2 Report

Comments and Suggestions for Authors

I thank the researchers for this nice work, which included synthesizing a large number of chemical compounds and testing them.

 Despite the quality of this work in terms of chemical synthesis, there are a number of shortcomings that the researcher needs to complete in order for the research to be considered acceptable for publication, including the following:

1-         The grammatical and syntax errors should be carefully checked throughout the text

2-         the work was confusing, and I was not able to understand the main goal of the research, whether the cells were used, or whether it had anything to do with the immunodeficiency virus. 

3-         The author should add the molecular dynamics and to be performed with a minimum trajectory of 100 ns.

Comments on the Quality of English Language

-         The grammatical and syntax errors should be carefully checked throughout the text

Author Response

Reviewer comments: I thank the researchers for this nice work, which included synthesizing a large number of chemical compounds and testing them.

Author response:  We thank the reviewer for their kind comments. We have endeavoured to revise and modify the manuscript which we hope will be acceptable. Our responses are provided in detail below.

Reviewer comments: Despite the quality of this work in terms of chemical synthesis, there are a number of shortcomings that the researcher needs to complete in order for the research to be considered acceptable for publication, including the following

Reviewer comment 1: The grammatical and syntax errors should be carefully checked throughout the text

Author response: We have carefully checked the document for grammar and syntax errors, and corrected the revised manuscript.

Reviewer comment 2: the work was confusing, and I was not able to understand the main goal of the research, whether the cells were used, or whether it had anything to do with the immunodeficiency virus.

Author response: We thank the reviewer for their comments. For clarification we provide the following short description of the work reported in the manuscript.

The goals of our research were as follows: We first explored the structural requirements for antiproliferative activity of a panel of 58 compounds containing the nitrovinylaryl pharmacophore in Burkitt’s lymphoma (BL) cell lines, a rare form of non-Hodgkin’s immune B-cell lymphoma. We then identified the most promising compounds for further evaluation in chronic lymphocytic leukaemia (CLL) which is a more common but related B-cell malignancy. This work describes the synthesis, characterisation and evaluation of a series of nitrostyrene, related nitrovinylanthracenes and structural modification of these anthracenes in these BL and CLL cell lines. Based on our previous work in this area of design of compounds with activity against BL and CLL, our objective was to identify potential lead compounds with potent activity in the CLL cell lines, for further preclinical progression.

The work is arranged in the following sections:

  • Synthetic routes to nitrostyrenes, nitrovinylanthracenes and structural modification of these anthracenes
  • Evaluation of panel of synthesised nitrostyrenes, nitrovinylanthracenes and structurally modified anthracenes in BL cell lines; SAR study and identification of potent lead compounds for progression
  • Selection of the most potent nitrostyrenes and nitrovinylanthracenes for evaluation in CLL cell lines; SAR study and identification of potent lead compounds for further study
  • Confirmation of proapoptotic activity of lead compounds for further preclinical evaluation
  • Confirmation of the nitrovinylaryl pharmacophore required for activity in CLL cells.

The immunodeficiency virus (HIV) is mentioned in the context of the Burkitt’s lymphoma cells which were used in the cell viability study. Burkitt’s lymphoma (BL) a rare form of non-Hodgkin’s immune B-cell lymphoma. Burkitt’s lymphoma is the second most common subtype of NHL that occurs in HIV-positive patients with a relatively high CD4 cell count.

The Burkitt’s lymphoma (BL) cell lines used in this study were: EBV-MUTU-1 (chemosensitive) and EBV+ DG-75 (chemoresistant), Ramos (BL, EBV negative) and Bjab (BL, EBV negative). BL is sub-classified into three clinical variants: endemic, sporadic, and immunodeficiency-related. The immunodeficiency-related BL subtype is most common in HIV/AIDS patients and accounts for 30 to 40% of AIDS-related non-Hodgkin’s lymphoma (NHL). The subtypes differ in geographical distribution and Epstein-Barr Virus (EBV) association. The endemic variant is associated with EBV infections, with 98% of endemic BL contain the EBV genome, while only 5–10% of sporadic BL are EBV-positive. EBV infections occur in 30–40% of the immunodeficiency-related BL subtypes associated with HIV, showing an association with EBV infections.

[see references: Berhan, A.; Bayleyegn, B.; Getaneh, Z. HIV/AIDS Associated Lymphoma: Review , Blood and Lymphatic Cancer: Targets and Therapy 2022:12 31–45; Guech-Ongey, M.; Simard, E.P.; Anderson, W.F.; Engels, E.A.; Bhatia, K.; Devesa, S.S.; Mbulaiteye, S.M. AIDS-related Burkitt lymphoma in the United States: What do age and CD4 lymphocyte patterns tell us about etiology and/or biology? Blood 2010, 116, 5600-5604.]

The CLL cell lines chosen for the study were as follows: HG-3 cell line was established from an in vitro EBV (Epstein Barr Virus) infection from an IGVH1–2 unmutated B1 lymphocyte origin CLL patient clone and is representative of poor patient prognosis [ref 78]. The PGA-1 cell line is a cell line established from leukemic B cells of a Caucasian male with CLL with a mutated IGVH1-2 and is representative of good patient prognosis [ref 79]

Reviewer comment 3: The author should add the molecular dynamics and to be performed with a minimum trajectory of 100 ns.

Author response: The series of nitrostyrene and nitrovinylanthracene compounds described in this paper were evaluated for antiproliferative activity in chronic lymphocytic leukaemia (CLL) and also in Burkitt’s lymphoma (BL) cell lines, a rare form of non-Hodgkin’s immune B-cell lymphoma. The compounds were designed from the tetracyclic scaffold structure of the antidepressant maprotiline and demonstrated antiproliferative and proapoptotic effects in the CLL cell lines. However, we do not have a detailed mechanism of action (MOA), i.e. their primary protein target is unknown. Without this knowledge we are unable to perform molecular dynamics calculations. The computational studies in this paper are focused on determining structural similarities between the novel compounds and maprotiline to gain insight into a possible related MOA.

Reviewer 3 Report

Comments and Suggestions for Authors

The current study explores the use of two scaffolds, (E)-nitrostyrene and (E)-9-(2-nitrovinyl)anthracene, to expand on the design, synthesis, and biological evaluation of related compounds. The work is extremely well organized and written, and the actual experimental output is as large as it is valuable.

I have a few minor issues I'd like to address:

 1.I noticed that compound 19a had previously been tested based on your previous work and the manuscript itself. Are all the other compounds that have been synthesized and tested biologically novel? If not, could you please specify which ones have previously been reported? If all of the compounds are new, I believe that should be highlighted as well.

2. In Scheme 2, the first reaction condition a) is defined as a two-step process with i) and ii); the same cation has "(i) (i) Zn dust, ammonia, 75 ℃, 4 h; (ii) HCl, isopropyl"; there are to (i) and then (ii). This is a little confusing; instead of so many "i"-s, you could use other annotations for multiple steps like a1, a2,... . You have left a minus or hyphen before 90°C in the same caption; please correct that as well.

3. The paragraph begins with a superscript "a" in line 652; was this part intended to be a footnote? If not, please remove it; if it was intended to be a footnote, I cannot find any "a"-s in figure 6. This issue also occurs in Figure 7/

4. You have the same starting footnote in table 2, but all I could find in the table were superscript "b"-s. There is no "a" in table 5, but the same footnote is present. Tables 6 and 7 have the same issue. Also, in the pdf version I'm reading, the 2D structures in table 6 overlap the borderlines of the table cells. Please double-check.

5. Regarding the viability method section, I don’t quite understand some details. Were all cells, including drug-untreated cells, treated with 1% ethanol? If that is the case, were there instances were ethanol induce significant viability reduction, because that is worth mentioning even if all results were normalized after 1% ethanol treated cells.

6. Please conduct statistical analysis on your viability results and add the p related significant values to your charts.

Author Response

Reviewer comment: The current study explores the use of two scaffolds, (E)-nitrostyrene and (E)-9-(2-nitrovinyl)anthracene, to expand on the design, synthesis, and biological evaluation of related compounds. The work is extremely well organized and written, and the actual experimental output is as large as it is valuable.

I have a few minor issues I'd like to address:

Author response: We thank the reviewer for their kind comments. We have endeavoured to revise and modify the manuscript which we hope will be acceptable. Our responses are provided in detail below

Reviewer comment 1. I noticed that compound 19a had previously been tested based on your previous work and the manuscript itself. Are all the other compounds that have been synthesized and tested biologically novel? If not, could you please specify which ones have previously been reported? If all of the compounds are new, I believe that should be highlighted as well.

Author response: The following compounds evaluated in this work are chemically novel and have not been previously reported: 14b, 19h, 19i, 19j, 19k,  19l, 19m, 20a, 20b, 22, 23a, 23b, 30b, 31, 33b, 33c, 33e. These compounds are now identified as “novel” or “previously unreported” in the revised text.

The chemical synthesis of the following compounds 11a-l, 13a-e, 14a, 14c-e, 14f-g, 15, 16, 18, 19a-g, 21, 23c, 24-28, 29a-d, 30a, 32a-e, 33a, 33d-f were previously reported in literature, (see page 43 of the manuscript text). These compounds have not been evaluated biologically previously, with the exception of 11a-l and 19a which we evaluated in BL (see references 39 and 41 in manuscript). Also see Supplementary Information for preparation and characterisation of these compounds [relevant literature references to the synthesis of these compounds are provided in the Supplementary Information Reference section].

Reviewer comment 2. In Scheme 2, the first reaction condition a) is defined as a two-step process with i) and ii); the same cation has "(i) (i) Zn dust, ammonia, 75 ℃, 4 h; (ii) HCl, isopropyl"; there are to (i) and then (ii). This is a little confusing; instead of so many "i"-s, you could use other annotations for multiple steps like a1, a2,... . You have left a minus or hyphen before 90°C in the same caption; please correct that as well.

Author response: The reaction conditions (a), (h) and (i) in the caption for Scheme 2 are now corrected as requested:

(a) (a1) for compounds 13a-c: RMgBr, toluene, reflux, 3h; (a2) HCl (20%), ice, (76-84%);

(h) N-Methylformanilide, n-BuLi, 90 oC, 50 min, (72%);

(i) (i1) Zn dust, ammonia, 75 oC, 4 h; (i2) HCl, isopropanol, 3 h, reflux, (57%);

90°C is also corrected in the caption for Scheme 2

Reviewer comment 3. The paragraph begins with a superscript "a" in line 652; was this part intended to be a footnote? If not, please remove it; if it was intended to be a footnote, I cannot find any "a"-s in figure 6. This issue also occurs in Figure 7/

Author response: In the starting footnote for Figures 6 and 7, The superscript “a” was a typographical error and is now removed

Reviewer comment 4. You have the same starting footnote in table 2, but all I could find in the table were superscript "b"-s. There is no "a" in table 5, but the same footnote is present. Tables 6 and 7 have the same issue. Also, in the pdf version I'm reading, the 2D structures in table 6 overlap the borderlines of the table cells. Please double-check.\

Author response:

The superscript “a” is now inserted in the title of Table 2

The superscript “a” is included in the Title for Table 5

The superscripts “a” and “b” are now inserted in the column headings and footnote for Table 6

The superscripts “a” and “b” are now inserted in the column headings and footnote for Table 7

We checked Table 6 again, and the 2D structures in Table 6 appear to be within the Table cells in the PDF version of the document submitted. We will confirm with the Editorial Office.

Reviewer comment 5. Regarding the viability method section, I don’t quite understand some details. Were all cells, including drug-untreated cells, treated with 1% ethanol? If that is the case, were there instances were ethanol induce significant viability reduction, because that is worth mentioning even if all results were normalized after 1% ethanol treated cells.

Author response: All cells including drug untreated cells were treated with the vehicle ethanol (1% v/v). Results were normalised to 100% viability after treatment with 1% ethanol and the ethanol treated wells (1 % v/v) were set at 100 % cell viability. Significant reduction in cell viability induced by the vehicle was not observed. The effects for untreated cells and vehicle treated cells are included in the Apoptosis Figure 11.

See text, Experimental section 4.3, lines 1527-1546 for details of  cell viability method.

Reviewer comment 6. Please conduct statistical analysis on your viability results and add the p related significant values to your charts.

Author response: We have updated the viability results to include statistical analysis.

Reviewer 4 Report

Comments and Suggestions for Authors

molecules-2710010

In this manuscript, the synthesis and antiproliferative activity of series of nitrovinylanthracenes and related nitrostyrenes are reported.

This is an interesting paper with a good description of both the chemical and biological methods used. The results and SARs are sound and will be of interest in scientists active in health sciences. In my opinion it could be accepted for publication in its current form. I have only two minor suggestions to be taken into account by the authors.

Line 160, please add “identified the related ( E ) - 9 -(2 -nitrovinyl)anthracene 19a, Figure …..”

Concerning already published compounds, some more informative related references should be provided and experimental data are not required in this case. For example: compound 13a has been described by Ikeda et al. in J. Amer. Chem. Soc. 1990, 112, 4650 – 4656, whereas the authors provide a 1923 reference (REF 4 in the supporting information file).

Author Response

Reviewer comment: In this manuscript, the synthesis and antiproliferative activity of series of nitrovinylanthracenes and related nitrostyrenes are reported.

This is an interesting paper with a good description of both the chemical and biological methods used. The results and SARs are sound and will be of interest in scientists active in health sciences. In my opinion it could be accepted for publication in its current form. I have only two minor suggestions to be taken into account by the authors.

Author response: We thank the reviewer for their kind comments. We have endeavoured to revise and modify the manuscript which we hope will be acceptable. Our responses are provided in detail below:

Reviewer comment: Line 160, please add “identified the related ( E ) - 9 -(2 -nitrovinyl)anthracene 19a, Figure …..”

Author response: The manuscript is now revised to contain the following : Line 160, “identified the related ( E ) - 9 -(2 -nitrovinyl)anthracene 19a, Scheme 3

Reviewer comment: Concerning already published compounds, some more informative related references should be provided and experimental data are not required in this case. For example: compound 13a has been described by Ikeda et al. in J. Amer. Chem. Soc. 1990, 112, 4650 – 4656, whereas the authors provide a 1923 reference (REF 4 in the supporting information file).

Author response: References provided in the Supplementary Information (Page 66-69) are now updated as follows:

In the Supplementary Information, references 2, 3, 4, 5, 6, 9, 10, 19 and 30 for original literature reports of compounds 13d, 13e, 13a, 13b, 13c, 14e, 15, 21 and 28 respectively have been replaced with more recent references, together with an additional reference 36 for compound 14d

Round 2

Reviewer 1 Report

Comments and Suggestions for Authors

Accept in present form.

Reviewer 2 Report

Comments and Suggestions for Authors

Authors have improved and corrected all the comments made by me in the first submission. Now, I think the article is suitable for publication